# A CRISPR-based rapid DNA repositioning strategy and the early intranuclear life of HSV-1

Juan Xiang[†], Chaoyang Fan[†], Hongchang Dong, Yilei Ma, Pei Xu*

The Centre for Infection and Immunity Studies, School of Medicine, Shenzhen Campus of Sun Yat-sen Univeristy, Sun Yat-sen University, Shenzhen, China

**\*For correspondence:**
xupei3@mail.sysu.edu.cn

[†]These authors contributed equally to this work

**Competing interest:** The authors declare that no competing interests exist.

**Abstract** The relative positions of viral DNA genomes to the host intranuclear environment play critical roles in determining virus fate. Recent advances in the application of chromosome conformation capture-based sequencing analysis (3 C technologies) have revealed valuable aspects of the spatiotemporal interplay of viral genomes with host chromosomes. However, to elucidate the causal relationship between the subnuclear localization of viral genomes and the pathogenic outcome of an infection, manipulative tools are needed. Rapid repositioning of viral DNAs to specific subnuclear compartments amid infection is a powerful approach to synchronize and interrogate this dynamically changing process in space and time. Herein, we report an inducible CRISPR-based two-component platform that relocates extrachromosomal DNA pieces (5 kb to 170 kb) to the **nu**clear **p**eriphery **in** minutes (CRISPR-nuPin). Based on this strategy, investigations of herpes simplex virus 1 (HSV-1), a prototypical member of the human herpesvirus family, revealed unprecedently reported insights into the early intranuclear life of the pathogen: (I) Viral genomes tethered to the nuclear periphery upon entry, compared with those freely infecting the nucleus, were wrapped around histones with increased suppressive modifications and subjected to stronger transcriptional silencing and prominent growth inhibition. (II) Relocating HSV-1 genomes at 1 hr post infection significantly promoted the transcription of viral genes, termed an 'Escaping' effect. (III) Early accumulation of ICP0 was a sufficient but not necessary condition for 'Escaping'. (IV) Subnuclear localization was only critical during early infection. Importantly, the CRISPR-nuPin tactic, in principle, is applicable to many other DNA viruses.

## Editor's evaluation

The authors present an ingenious approach to investigate the effects of cellular location on herpes virus replication. By infecting cells that express a fusion protein between Cas9 and emerin, a nuclear matrix protein, the addition of guide RNAs that target the herpes virus DNA genome quickly localize it to the nuclear membrane. Initial inhibition of viral gene expression, followed by release of that inhibition in just one hour, was observed. This is an interesting approach to test the importance of localization within the nucleus that is reveals both an effect of 'tethering' and a pathway for escape of that tethering, using a method relevant to other viral and non-viral nucleic acids.

## Introduction

Cellular chromosomes are not randomly positioned in the nucleus, and the dynamic organization of intranuclear DNA in space and time plays a fundamental role in regulating the cellular transcriptome. Recent innovations in mapping and manipulative technologies have boosted scientific enthusiasm to address the cause-effect relationship between gene transcriptional regulation and its relative

intranuclear locus on the four-dimensional scale (*Schoenfelder and Fraser, 2019*; *Marchal et al., 2019*; *Wang et al., 2021*). The recognition of the complexity and the critical role played by higher order genomic organization in transcriptional regulation has generated immense interest in the field of virology, which holds that DNA viruses, the co-evolved parasites of host machinery, are interrelated to and/or exploit 3D genome organization at the cell level. Currently, evidence supporting this concept is mostly from descriptive experiments identifying and/or characterizing host genomic hot spots in association with viral genomes using immunofluorescence staining and fluorescent in situ hybridization (FISH) or, more recently, by chromosome conformation capture (3 C)-based methods (*Yue et al., 2021*; *Tang et al., 2021*; *Xiao et al., 2021*; *Yang et al., 2020*; *Hensel et al., 2018*). Few studies have examined the causal relationship underlying intranuclear host-virus spatial correlations, partly due to a lack of appropriate probes. Herein, we took advantage of the recent advances in CRISPR-based genome re-organization tools and constructed an inducible two-component CRISPR-nuPin platform that enabled repositioning of extrachromosomal DNA pieces to the nuclear edge within minutes. Furthermore, we used the system to interrogate the early intranuclear life of HSV-1, a prototypic medium-sized DNA virus, in space and time.

HSV-1, a member of subfamily *Alphaherpesvirinae*, family *Herpesviridae*, is a double stranded DNA virus of approximately 153 kb in size that infects over 66.7% of the population of 0–49 year-olds globally (*James et al., 2020*). A mature, infectious HSV-1 virion consists of a lipid envelope, the tegument layer and an icosahedral capsid wrapping around a single copy of the viral genome. HSV-1 uses the strongest molecular motor known (the portal vertex) to pack its negatively charged genome into a relatively tiny space, resulting in tens of atmospheres of pressure inside the capsid (*Bauer et al., 2013*; *Brandariz-Nuñez et al., 2019*; *McElwee et al., 2018*; *Liu et al., 2019*). The viral capsid docks itself to the nuclear pore complex (NPC), and the internal pressure provides an essential driving force for viral genome release and transportation through the NPC in a rod-like structure (*Bauer et al., 2013*; *Brandariz-Nuñez et al., 2019*; *Brandariz-Nuñez et al., 2020*; *Ojala et al., 2000*; *Newcomb et al., 2009*; *Shahin et al., 2006*). After genome ejection into the nucleus, HSV-1 initiates an active lytic replication cycle in almost every susceptible cell line in vitro, and three sets of viral genes, termed immediate early (α), early (β) and late genes (γ$_1$, γ$_2$), are to be expressed in a tightly regulated chronological order. The immediate events post nuclear entry of viral genomes center around a raging competition between the urge of the virus to express its transcripts and the repressive forces from the host to silence these exogenous DNAs. Abundant evidence shows that nuclear HSV-1 genomes are immediately sensed, assembled with histones, associated with de novo-formed nuclear bodies (PML bodies), and loaded with host proteins, including repressor complexes, to restrict viral transcription and defend against infection (*Roizman et al., 2005*; *Burkham et al., 1998*; *Knipe and Cliffe, 2008*; *Everett et al., 2004*). To counteract host intrinsic immunity, HSV-1 viral tegument protein αTIF (VP16) and its associated complex are transported into the nucleus independently of the viral capsid and initiate efficient transcription of α genes, including infected cell protein 0 (ICP0) (*Triezenberg et al., 1988*; *Stern and Herr, 1991*; *Babb et al., 2001*; *Douville et al., 1995*; *Cabral et al., 2018*). ICP0 is a 775 amino acid protein, multifunctional E3 ligase and promiscuous transactivator essential for the expression of post-α genes (β and γ genes) at a low multiplicity of infection (*Hagglund and Roizman, 2004*). The protein executes multiple transcription-relevant functions: the dispersal of PML bodies through proteasome-dependent degradation of PML and Sp100, inactivation of the HDAC1/CoREST/LSD/REST repressor complex, recruitment of CLOCK H3/H4 acetyltransferases to activate viral transcription, removal of heterochromatin packaging of viral genomes and reduction of overall histone association of HSV-1 DNAs (*Roizman et al., 2005*; *Gu and Roizman, 2009*; *Gu and Roizman, 2007*; *Cliffe and Knipe, 2008*; *Kalamvoki and Roizman, 2010*; *Kalamvoki and Roizman, 2011*; *Lee et al., 2016*). An intriguing but blurred aspect of these elegantly executed molecular tasks is the precise sequential order of these events and their relationship with the diverse spatial localizations of the viral genomes in the host nuclei.

Herein, utilizing the CRISPR-nuPin strategy, we took a very first step addressing the complicated and heterogeneous host-virus interactions on a four-dimensional scale and revealed several intriguing aspects of the early intranuclear life of HSV-1. In summary, we report that (I) the CRISPR-nuPin strategy is a two-component platform consisting only of a dCas9-emerin fusion protein and sgRNAs, which is smaller and simpler than most of the previously reported genome repositioning techniques. (II) CRISPR-nuPin drove relocation of extrachromosomal DNA pieces up to 170 kb to the inner nuclear

margin efficiently and promptly upon electroporation of sgRNAs. CRISPR-nuPin can reposition target DNAs via a single sgRNA targeting site and thus has wide application to various DNA viruses replicating in the nucleus. (III) HSV-1 genomes deposited to the nuclear edge immediately upon entry were subjected to stringent transcriptional silencing and major growth inhibition, up to a multiplicity of infection (MOI) of 20. (IV) dislodging HSV-1 genomes from their originally occupied spatial niches to the nuclear margin at 1 hpi significantly promoted viral gene expression and virus production ("Escaping"). (V) early accumulation of ICP0 protein was a sufficient but not necessary condition in "Escaping". Interestingly, subnuclear localization bias was only detected during the early period of infection. Shortly after nuclear entry, HSV-1 genomes are no longer sensitive to subnuclear localization changes.

## Results

### A two-component CRISPR-nuPin platform that mediates nuclear edging of extrachromosomal DNAs within minutes

The life cycle of many DNA viruses is in the range of hours (*Roizman and Knipe, 2001*). Therefore, the probing tool to interrogate the spatial interactions between viral genomes and the host subnuclear environment needs to act quickly. To simplify the DNA repositioning system and avoid the possible time lag introduced by a second layer of regulation (e.g. the abscisic acid inducible ABI-PYL1 system used by CRISPR-GO and CRISPR-based CLOuD9 or the CIBN-CRY2 system used by CRISPR-based LADL; *Morgan et al., 2017*; *Wang et al., 2018*; *Kim et al., 2019*), we tested whether a two-component strategy, including only a subnuclear locus-anchored dcas9 and a sgRNA, was able to mediate inducible and fleet relocation of viral DNA genomes (*Figure 1A*). To achieve nuclear periphery docking, Flag-emerin-TEV (TEV protease recognition sequence)-GFP was fused to the C-terminus of *Streptococcus pyogenes* dCas9 (D10A&H840A), resulting in a fusion protein with a predicted size of 230 kDa (dCas9-emerin; *Figure 1B*, *Figure 1—source data 1 and 2*, and *Figure 1—figure supplement 1A*). Insertion of a nuclear localization signal (NLS) at the N- and C-termini of dCas9 enabled nuclear importation of the majority of dCas9-emerin, while Flag-tagged emerin localized both in the nucleus and cytoplasm (*Figure 1B*, *Figure 1—source data 1 and 2*). We then constructed a doxycycline (DOX)-inducible dCas9-emerin-expressing HEp-2 cell line (dCas9-emerin) using a lentiviral transduction system and a HEp-2 cell line expressing only dcas9 protein, serving as a control cell line (dCas9; *Figure 1—figure supplement 1B*, *Figure 1—figure supplement 1—source data 1–3*). Successful induction of the dCas9-emerin fusion protein in dCas9-emerin cells by DOX was confirmed by immunoblotting and IF (*Figure 1—figure supplement 1C, D*, *Figure 1—figure supplement 1—source data 4–6*). Colocalization of the anti-Flag antibody recognized epitope with the nuclear envelope (represented by NUP98 staining) along the outer edge of the DAPI-stained nuclear area indicated that the dcas9-emerin fusion protein was properly anchored to the inner nuclear membrane (*Figure 1C, D*, *Figure 1—source data 3 and 4* and *Figure 1—figure supplement 1D–F*, *Figure 1—figure supplement 1—source data 6–8*). Note that in a portion of dcas9-emerin cells, the fusion protein was expressed abundantly, and cytosolic staining was detected (*Figure 1—figure supplement 1D, F*). The addition of DOX to the culturing medium had no detectable cytotoxicity on HEp-2 cells (*Figure 1—figure supplement 1G*, *Figure 1—figure supplement 1—source data 9*).

To test the efficiency of CRISPR-nuPin, dCas9-emerin cells were electroporated with a 5 kb plasmid (*Figure 1—figure supplement 1H,I*, *Figure 1—figure supplement 1—source data 9 and 10*) or a 170 kb BAC plasmid containing the HSV-1 genome (BAC) (*Figure 1E and F*, *Figure 1—source data 5 and 6*) and again with their respective targeting sgRNAs or control sgRNA (ctrl sgRNA) 24 hr later. Representative 3D images of the nucleus at the indicated time points post sgRNA electroporation (upper panel) and their 3D reconstruction images (bottom panel) are shown in *Figure 1—figure supplement 1H* and *Figure 1E*. More than 500 spots of each sampling group were counted, and their distribution pattern, either in the nuclear margin or in the inner nuclear space, was plotted. In brief, when a FISH-stained spot was partially exposed outside of or completely exposed but still adjacent to the DAPI region after 3D reconstruction, it was categorized as a border-located spot. When a FISH-stained spot was completely immersed in the DAPI region, it was counted as an inner nuclear space localized spot. FISH spots that were not associated with the DAPI area were dismissed during counting. A 3D cross-section image showing the relative position of nuclear edge spots to the

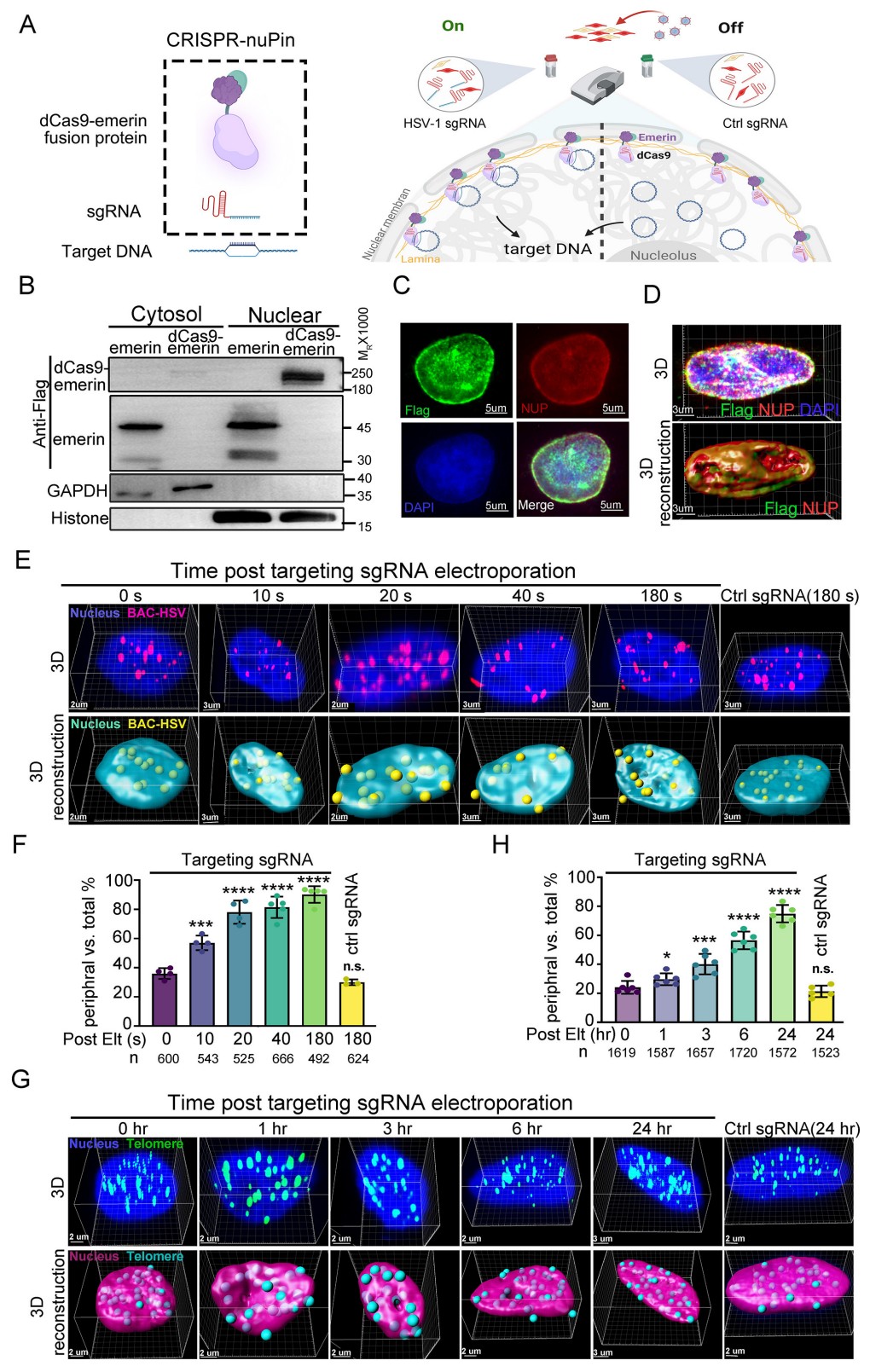

**Figure 1.** The CRISPR-nuPin system. (**A**) Schematics of the CRISPR-nuPin system. (**B**) Subcellular localization of transiently expressed Flag-dCas9-emerin and Flag-emerin in HEK293T cells. (**C–D**) dCas9-emerin cells were treated with doxycycline (+DOX), and the intracellular distribution of dCas9-emerin protein was examined by immunofluorescence staining with anti-Flag and anti-NUP98 (NUP) antibodies. Representative images scanned by

*Figure 1 continued on next page*

*Figure 1 continued*

confocal microscopy and 3D reconstructed are shown in **D**. (**E–G**) BAC-HSV-1 plasmids containing dCas9-emerin cells (+DOX) were electroporated with sgRNA targeting HSV-1 or ctrl sgRNA (**E, F**), or dCas9-emerin cells (+DOX) only were electroporated with telomere-targeting sgRNA or ctrl sgRNA (**G, H**). At the indicated time points post sgRNA electroporation, cells were fixed, and the intranuclear BAC or telomeres were stained by FISH and scanned under a confocal microscope. Representative images (upper panel) and their 3D reconstructions (bottom panel) are shown in (**E and G**). The percentage of the BACs or telomeres at the nuclear edge vs total intranuclear stains at each time point was calculated and plotted in **F** (BAC) and **H** (telomere), respectively. The total number of counted dots of each sampling group is shown as n at the bottom of (**F and H**). In panel (**F**) an average of approximately 30–50 nuclei were counted. In panel (**H**) approximately 27–30 nuclei were counted to generate the respective datasets. Data is shown as mean ± SD, n = 3, and P values are calculated using the Student's t-test. n.s. represents not significant, p>0.05, '*' represents p≤0.05, '**' represents p≤0.01, '***' represents p≤0.001, and '****' represents p≤0.0001.

The online version of this article includes the following source data and figure supplement(s) for figure 1:

**Source data 1.** Source files for western blot, statistic data and images in *Figure 1*.

**Source data 2.** Uncropped western blot with labels in *Figure 1B*.

**Source data 3.** Source files for processed images in *Figure 1C*.

**Source data 4.** Source files for processed images in *Figure 1D*.

**Source data 5.** Source files for processed images in *Figure 1E*.

**Source data 6.** Source files for original data collected for statistical analysis in *Figure 1F, H*.

**Source data 7.** Source files for processed images in *Figure 1F*.

**Figure supplement 1.** Characterization of the CRISPR-nuPin system.

**Figure supplement 1—source data 1.** Source files for western blot, statiscic data and images in *Figure 1—figure supplement 1*.

**Figure supplement 1—source data 2.** Source files for uncropped western blot in *Figure 1—figure supplement 1B*.

**Figure supplement 1—source data 3.** Source files for uncropped western blot with labels in *Figure 1—figure supplement 1B*.

**Figure supplement 1—source data 4.** Source files for uncropped western blot in *Figure 1—figure supplement 1C*.

**Figure supplement 1—source data 5.** Source files for uncropped western blot with labels in *Figure 1—figure supplement 1B*.

**Figure supplement 1—source data 6.** Source files for processed images in *Figure 1—figure supplement 1D*.

**Figure supplement 1—source data 7.** Source files for processed images in *Figure 1—figure supplement 1E*.

**Figure supplement 1—source data 8.** Source files for processed images in Figure 1-figure supplement 1 F.

**Figure supplement 1—source data 9.** Source files for original data collected for statistical analysis in Figure 1-figure supplement 1 G and I.

**Figure supplement 1—source data 10.** Source files for processed images in *Figure 1—figure supplement 1H*.

**Figure supplement 1—source data 11.** Source files for processed images in *Figure 1—figure supplement 1J*.

**Figure supplement 1—source data 12.** Source files for processed images in *Figure 1—figure supplement 1K*.

DAPI-stained area is presented in *Figure 1—figure supplement 1J* (*Figure 1—figure supplement 1—source data 11*). This concept applies to all the following experiments concerning the intranuclear localization of HSV-1 viral DNAs (the exact descriptive terms used include nuclear edge, nuclear margin, nuclear periphery, in proximity to the inner nuclear membrane, etc.) in this study. As shown in *Figure 1F*, 10–20% of BAC DNAs started to be relocated away from the nuclear interior at 10 s post sgRNA electroporation, and more than 80% of BAC DNAs were relocalized to the nuclear edge after 3 min of sgRNA electroporation, while introduction of ctrl sgRNA barely had any detectable impact.

To test the repositioning efficiency of CRISPR-nuPin on the host genomic locus, dCas9-emerin cells were electroporated with sgRNA targeting the telomere region (*Figure 1G and H*, *Figure 1—source data 6 and 7*). To determine the subnuclear localization of telomeres, the nuclear margin was restrained to the space between the NUP98-stained nuclear envelope and the DAPI-stained area (a 3D cross-section image showing the relative position of nuclear edge spots to NUP98- and DAPI-stained

areas is presented in *Figure 1—figure supplement 1K*, *Figure 1—figure supplement 1—source data 12*). At 3 hr post sgRNA induction, approximately 38% of cellular telomeres were detected bound to the nuclear edge compared to approximately 25% at time 0 hr and 21% for ctrl sgRNA. Our results indicate that the CRISPR-nuPin system is capable of repositioning of extrachromosomal DNA of up to 170 kb in size to the nuclear periphery within minutes. Note that a single dot in the 3D reconstruction images represents single or multiple plasmids or BACs.

## HSV-1 genomes positioned to the nuclear margin were subjected to transcriptional repression

To replicate, most DNA viruses need to eject their genetic materials into the host nuclei and initiate viral genome transcription. It is conceivable that viral genomes encounter highly heterogenic micro-environments upon entry, even within the same nucleus. In fact, approximately 25% of the HSV-1 genomes are in proximity to the inner nuclear membrane during a natural infection, and the rest penetrate further into the nuclear interior (*Figure 2—figure supplement 1A*, *Figure 2—figure supplement 1—source data 1*). The distance from the center of a nuclear edge-localized HSV-1 FISH spot to the outline of DAPI was within 0.9 µm (median value 0.44 µm and 0.44 µm on average), and the diameter for an HSV-1 FISH spot was 0.59 µm on average under the experimental settings in the current study (*Figure 2—figure supplement 1B*, *Figure 2—figure supplement 1—source data 1 and 2*). We then asked if viral DNAs preferred the nuclear interior to the nuclear edge for infection. To this end, 5 sgRNAs of different repeating numbers of targeting sites within the HSV-1 genome were designed to enable relocation of HSV-1 genomes in the CRISPR-nuPin platform. Cautions were made to avoid protein coding regions of the virus (*Figure 2—figure supplement 1C*). All sgRNAs were able to mediate significant inhibition of HSV-1 in dCas9-emerin cells (*Figure 2—figure supplement 1*, *Figure 2—figure supplement 1—source data 1*). Since sgRNA2 binds to one site at the coding region for the 8.3 kb latency-associated transcript (LAT) precursor of HSV-1, which plays assumed limited role in HSV-1 lytic replication (*Garber et al., 1997*), and another site in the terminal repeat short regions (TRs), it was used to mediate BAC and HSV-1 genome repositioning in the rest of the experiments in this study (designated HSV-1 sgRNA). To exclude the effect of nonspecific factors such as dCas9-sgRNA binding to viral genomes or sgRNA-induced innate immune responses, HEp-2 cells expressing GFP, dCas9 or dCas9-emerin were transfected with a plasmid encoding ctrl sgRNA or HSV-1 sgRNA. The cells showed no essential differences in supporting HSV-1 infection, except for those producing both the dCas9-emerin fusion protein and the HSV-1 sgRNA (*Figure 2—figure supplement 1E,F,G*, *Figure 2—figure supplement 1—source data 1*). The single-cycle growth kinetics of HSV-1 were examined in dCas9-emerin cells in the presence of ctrl sgRNA or HSV-1 sgRNA. In dCas9-emerin cells expressing the HSV-1 sgRNA, viral genomes were guided to the nuclear edge compared to the ctrl sgRNA group, and tethering viral genomes to the inner nuclear membrane led to reduced virus production at all time points, including the plateau phase (*Figure 2A and B*, *Figure 2—source data 1 and 2*). Increasing the multiplicity of infection up to 20 infectious virions per cell did not help the virus overcome the plight (*Figure 2C*, *Figure 2—source data 1*, *Figure 2—figure supplement 1H*, *Figure 2—figure supplement 1—source data 1*). To exclude the possibility that CRISPR-nuPin relocated the viral genomes outside of the nucleus to places like the ER, cells were fractioned into subcellular compartments (cytosol, nucleus and membranous organelles), and the intranuclear viral genome level was examined. No consistent differences were detected between the HSV-1 sgRNA group and the ctrl sgRNA group across independent experiments (*Figure 2—figure supplement 1I*, *Figure 2—figure supplement 1—source data 1; 4 and 5*).

HSV-1 transcribes three sets of viral genes in chronological order (*Dalrymple et al., 1985*; *Lu and Misra, 2000*; *Gruffat et al., 2016*). Intriguingly, in the host nuclei with the viral genomes positioned to the nuclear edge, the mRNA levels of the representative viral genes of the three classes declined sharply to a significantly lower basal level before the transcription was fully activated. Although the launch of active transcription of ICP27, a representative α gene of HSV-1, in HSV-1 sgRNA-transfected dCas9-emerin cells was delayed by 30 min compared to that in the ctrl sgRNA group, the transcription rates were later comparable. The total amount of ICP27 mRNA synthesized by the nuclear-edged HSV-1 remained 10-fold lower than that synthesized by HSV-1 freely infecting the nuclei (*Figure 2D*, *Figure 2—source data 1*). Similar patterns were observed for the representative β (TK) and γ (VP16) genes with a slightly longer delay of the onset of full-speed transcription (*Figure 2D*, *Figure 2—source*

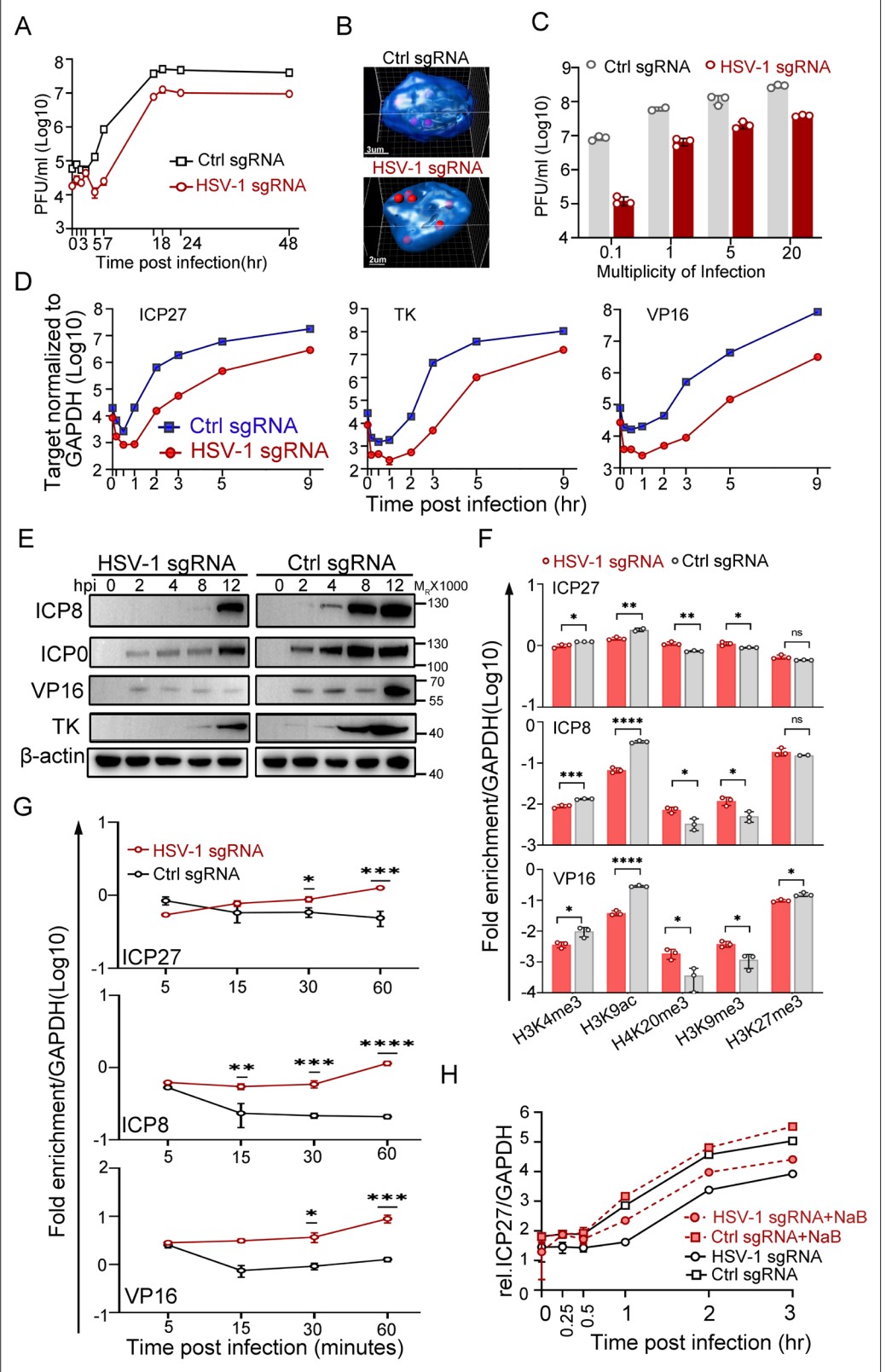

**Figure 2.** Transcription from the nuclear edge-localized HSV-1 genomes upon entry was strongly inhibited. (**A–F**) dCas9-emerin cells transfected with HSV-1 sgRNA or ctrl sgRNA for 24 hr were infected with HSV-1 at an MOI of 1 (2 hr on ice) (**A, F, G, H**), an MOI of 5 (**B, D, E**), or the indicated MOIs (**C**). (**A**) Single-cycle growth kinetics of HSV-1 titrated by plaque assay, data shown are mean ± SD, n = 3. (**B**) Cells were fixed at 3 hpi and stained for HSV-1

*Figure 2 continued on next page*

*Figure 2 continued*

genomes by FISH. The 3D reconstruction of the image is shown at the bottom. (**C**) Cell-associated HSV-1 viruses were titrated at 12 hpi, data shown are mean ± SD, n = 3. (**D**) Total RNA was extracted at 0, 0.17, 0.5, 1, 2, 3, 5 and 9 hpi, and the mRNA levels of ICP27, TK, ICP0 and VP16 were measured by qPCR (0.17 and 0.5 hpi labels were omitted), data shown are mean ± SD, n = 3. (**E**) The protein levels of ICP0, ICP8, TK, and VP16 at the indicated time points post infection were examined by immunoblotting. β-actin served as a loading control. (**F**) ChIP assays of differentially modified histones on the promoters of ICP27, ICP8 and VP16 of HSV-1 in HSV-1 sgRNA- or ctrl sgRNA-expressing dCas9-emerin cells at 0.5 hpi were performed using antibodies specific for H3K4me3, H3K9ac, H4K20me3, H3K9me3 and H3K27me3. Data is shown as mean ± SD, n = 3, and P values are calculated using the Student's t-test. n.s. represents not significant, p>0.05, '*' represents p≤0.05, '**' represents p≤0.01, '***' represents ≤0.001, and '****' represents p≤0.0001. (**G**) ChIP assays as in **F** were performed at the indicated time points post infection using an antibody against H3K9me3. For (**F and G**) the results were expressed as enrichment (fold) by comparing the fraction of viral promoters immunoprecipitated by the indicated antibodies to the fraction of GAPDH immunoprecipitated in the same reaction. Data is shown as mean ± SD, n = 3, and P values are calculated using the Student's t-test. '*' represents p≤0.05, '**' represents p≤0.01, '***' represents ≤0.001, and '****' represents p≤0.0001. (**H**) HSV-1 sgRNA- or ctrl sgRNA-expressing dCas9-emerin cells were treated with sodium butyrate (NaB) for 5 hr before infection (red lines) or not treated (black lines). The mRNA level of ICP27 at the indicated time points was measured by qPCR, data is shwon as mean ± SD, n = 3.

The online version of this article includes the following source data and figure supplement(s) for figure 2:

**Source data 1.** Source files for western blot, statiscic data and images in *Figure 2*.

**Source data 2.** Source files for processed images in *Figure 2B*.

**Source data 3.** Source files for uncropped western blot in *Figure 2E*.

**Source data 4.** Source files for uncropped western blot with relevant labels in *Figure 2E*.

**Figure supplement 1.** Nuclear edging of the viral genomes upon entry suppressed HSV-1 infection.

**Figure supplement 1—source data 1.** Source files for western blot, statiscic data and images in *Figure 2—figure supplement 1*.

**Figure supplement 1—source data 2.** Source files for processed images in *Figure 2—figure supplement 1B*.

**Figure supplement 1—source data 3.** Source files for processed images in *Figure 2—figure supplement 1G*.

**Figure supplement 1—source data 4.** Source files for uncropped western blot in *Figure 2—figure supplement 1L*.

**Figure supplement 1—source data 5.** Source files for uncropped western blot with labels in *Figure 2—figure supplement 1I*.

**Figure supplement 1—source data 6.** Source files for processed images in *Figure 2—figure supplement 1J*.

*data 1*). Postponed transcriptional onset led to delayed accumulation of viral proteins, including ICP8, ICP0, VP16 and TK (*Figure 2E*, *Figure 2—source data 3 and 4*), and decreased replication compartment-forming efficiency (*Figure 2—figure supplement 1J,K*, *Figure 2—figure supplement 1—source data 1 and 6*; *Xu and Roizman, 2017*). The observations in this set of experiments imply that deposition of HSV-1 genomes to the nuclear border led to decreased initial basal transcription, postponed full-speed transcription onset of viral gene cascades, severely delayed viral protein accumulation, inefficient replication of the virus, and ultimately reduced virus production.

The nuclear periphery is known to be decorated with heterochromatin (*Dekker et al., 2013*; *Dekker et al., 2002*; *Dixon et al., 2012*; *Lieberman-Aiden et al., 2009*). To investigate whether viral genomes at the nuclear edge were subjected to histone-mediated transcriptional suppression, ChIP-qPCR examining the histone modification patterns at the promoter regions of HSV-1 α, β and γ genes was performed (antibody specificity test over IgG is shown in *Figure 2—figure supplement 1L*, *Figure 2—figure supplement 1—source data 1*). Intriguingly, while histone packaging of the incoming viral genomes was not affected by their subnuclear positions at the time investigated (*Figure 2—figure supplement 1M*, *Figure 2—figure supplement 1—source data 1*), the nucleosomes wrapping HSV-1 DNAs appeared to be differentially modified. At 0.5 hpi, for viral genomes positioned to the nuclear edge, the promoters of representative viral genes showed significantly decreased association with active histone modifications (H3K4me3 and H3K9ac) and increased association with repressively modified histones (H4K20me3, H3K9me3) compared with those in the ctrl sgRNA group (*Figure 2F*, *Figure 2—source data 1*). Furthermore, the H3K9me3 association level

on viral genomes at the nuclear periphery progressively increased over time during the first hour of infection and was significantly higher than its level on HSV-1 DNA freely infecting the nuclei at 30- and 60 min post infection (*Figure 2G*, *Figure 2—source data 1*). Whole-cell inhibition of histone deacetylase (HDAC) activity with sodium butyrate (NaB; *Figure 2H*, *Figure 2—source data 1*) or other commonly used HDAC inhibitors (HDACi; *Figure 2—figure supplement 1N*, *Figure 2—figure supplement 1—source data 1*) showed that HDACi treatment could effectively derepress the nuclear edged-viral genomes. However, it should be noted that HDACi treatment only partially rescues the nuclear margining-imposed transcriptional suppression of HSV-1, hinting at the existence of other transcriptional suppression factors.

These results, in all, lead to an intriguing inference that intranuclear viral genomes face diversified fates upon entry and that those wandering in the nuclear edge area are disadvantaged during host–virus competition in lytic infection.

## Nuclear edging of the replicating HSV-1 genomes at 1 hpi promoted virus transcription, termed an 'Escaping' effect

HSV-1 genomes encounter waves of host restrictive responses once in the nucleus (*Merkl et al., 2018*; *Xu et al., 2016*). We hypothesized that during the process of intranuclear HSV-1 DNAs being sensed, recognized, marked, and eventually targeted (transcription suppression or degraded) by the host defense system, dragging away viral genomes from their original spots could interfere with these tightly regulated events and help virus escape. In this section, a series of experiments were conducted to test this hypothesis and probe the chronological order of host antiviral responses.

In several studies, it was reported that during HSV-1 infection, the virus disrupts the nuclear envelope and modifies and redistributes emerin via interaction with the viral helicase ICP8 (*Sagou et al., 2010*; *Leach et al., 2007*; *Morris et al., 2007*; *Maric et al., 2014*). We therefore examined the distribution pattern of dcas9-emerin, NUP98 and emerin under our experimental conditions to address the reliability and applicability of CRISPR-nuPin. At up to 6 hpi (MOI of 5), the majority of emerin localized at the nuclear envelope (*Figure 3—figure supplement 1A,B*, *Figure 3—figure supplement 1—source data 1 and 2*). At 3 hpi and 5 hpi, the nuclear envelope showed no observable disruption under a confocal microscope, and the distribution pattern of Flag-tagged dcas9-emerin was not distinguishable from that in the mock-infected cells (*Figure 3A*, *Figure 3—source data 1*, and *Figure 3—figure supplement 1C*, *Figure 3—figure supplement 1—source data 3*). In addition, cells showed clear nuclear envelope disruption combined with aberrant localization patterns of Flag-dcas9-emerin at 16 hpi, which agrees with previous reports that virus-induced modifications to emerin and the nuclear envelope occur in the late postinfection period (*Figure 3—figure supplement 1C*, *Figure 3—figure supplement 1—source data 3*; *Sagou et al., 2010*; *Leach et al., 2007*; *Morris et al., 2007*; *Maric et al., 2014*). In summary, the nuclear envelope remains intact, and dcas9-emerin remains localized on the nuclear envelope for at least the first five hours after infection at an MOI of 5 or less. Thus, CRISPR-nuPin is well suited for the following investigations.

First, dCas9-emerin cells infected with HSV-1 at an MOI of 1 were electroporated with HSV-1 sgRNA or ctrl sgRNA at 0 (immediately after 2 hr of virus inoculation on ice), 0.5, 1, and 2 hpi (*Figure 3B*). Examination of the subnuclear localizations of HSV-1 genomes at 5 min post sgRNA electroporation showed that CRISPR-nuPin mediated prompt and highly effective intranuclear relocation of viral genomes during active infection (*Figure 3C and D*, *Figure 3—source data 2 and 3*). Titration of the cell-associated viruses showed that the inhibitory effect imposed by nuclear edging of HSV-1 genomes was time sensitive and only effective during the very early and short period of time (approximately 0.5–1 hr under the current experimental conditions; *Figure 3E*, *Figure 3—source data 3*). An intriguing finding was that the virus yield was significantly increased when HSV-1 genomes were dislocated from their initial intranuclear position to the nuclear rim at 1 hpi (*Figure 3E*, *Figure 3—source data 3*). Tracking the virus production at multiple time points during infection confirmed that repositioning HSV-1 genomes to the nuclear periphery at 1 hpi switched the virus production level to a significantly higher level as early as 4 hpi (*Figure 3F*, *Figure 3—source data 3*). We thus designated this promotive effect mediated by nuclear edging of HSV-1 genomes at 1–1.5 hpi as 'Escaping'. 'Escaping' was not detected in dcas9 cells (*Figure 3—figure supplement 1D*, *Figure 3—figure supplement 1—source data 4*).

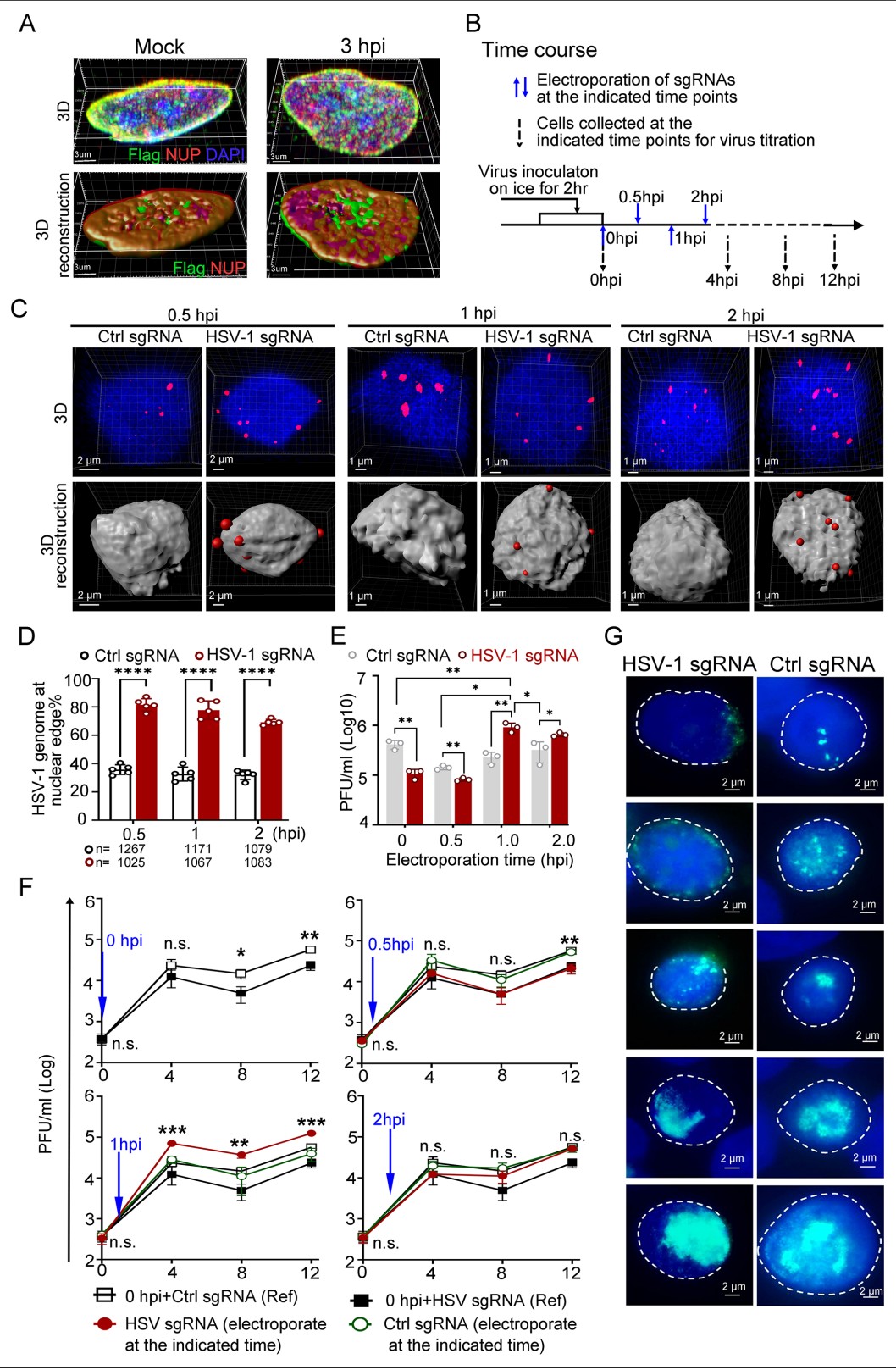

**Figure 3.** Dislodging HSV-1 genomes to the nuclear edge at 1 hpi promoted virus infection, termed 'Escaping'. (**A**) dCas9 cells (+DOX) mock-infected or infected with HSV-1 were fixed at 3 hpi, and the intracellular distribution of dCas9-emerin and NUP98 was stained with anti-Flag (green) and anti-NUP98 (NUP) (red) antibodies. (**B**) Schematics of the following experiments. dCas9-emerin cells were infected (2 hr on ice) with HSV-1 at an MOI of 1 and

*Figure 3 continued on next page*

*Figure 3 continued*

electroporated with HSV-1 sgRNA or ctrl sgRNA at 0 (immediately post virus inoculation), 0.5, 1 and 2 hpi. (**C–D**) At 5 min post sgRNA electroporation at the indicated time, the cells were fixed, and the intranuclear HSV-1 genomes were stained by FISH (red). Representative confocal images (upper panel) and their 3D reconstructions (3D reconstruction shows only the surface of the DAPI-stained area) are shown in **C**. More than 1000 total FISH-stained dots from approximately 155–250 nuclei were counted in each sampling group (n represents the total number of counted dots in each experimental group) The percentage of marginalized HSV-1 genomes vs total intranuclear counts is plotted in **D**. Data is shown as mean ± SD, n = 3, and P values are calculated using the Student's t-test. '****' represents p≤0.0001. (**E**) At 12 hpi, cell-associated viruses were titrated. Data is shown as mean ± SD, n = 3, and P values are calculated using the Student's t-test. '*' represents p≤0.05, '**' represents p≤0.01. (**F**) Cell-associated virus yields during the first 12 hr post infection were examined. dCas9-emerin cells electroporated with ctrl sgRNA (empty rectangle) or HSV-1 sgRNA (filled rectangle) at 0 hpi served as a reference in the three charts in **F**. The blue arrow indicates the time of sgRNA electroporation. The red filled cycle represents the virus yield in cells that received HSV-1 sgRNA, and the green empty cycle represents the virus yield in cells that received ctrl sgRNA. Data is shown as mean ± SD, n = 3, and P values are calculated using the Student's t-test. n.s. represents not significant, p>0.05, '*' represents p≤0.05, '**' represents p≤0.01, and '***' represents ≤0.001. (**G**) At 6 hpi, cells were fixed and stained with anti-ICP8 (green). Representative images of all stages of replication center formation are shown.

The online version of this article includes the following source data and figure supplement(s) for figure 3:

**Source data 1.** Source files for western blot, statiscic data and images in *Figure 3*.

**Source data 2.** Source files for processed images in *Figure 3C*.

**Source data 3.** Source files for original data collected for statistical analysis and for virus titers in *Figure 3D–F*.

**Source data 4.** Source files for processed images in *Figure 3G*.

**Figure supplement 1.** Localization of emerin during HSV-1 infection.

**Figure supplement 1—source data 1.** Source files for western blot, statiscic data and images in *Figure 3—figure supplement 1*.

**Figure supplement 1—source data 2.** Source files for processed images in *Figure 3—figure supplement 1B*.

**Figure supplement 1—source data 3.** Source files for processed images in *Figure 3—figure supplement 1C*.

**Figure supplement 1—source data 4.** Source files for original virus titers in *Figure 3—figure supplement 1D*.

One possible explanation for 'Escaping' was that placing viral genomes to the nuclear rim increases their chance and frequency of meeting the newly and cytosolically translated viral proteins passing through the NPC. To test this hypothesis, HSV-1-infected cells that received viral-specific sgRNA or ctrl sgRNA at 1 hpi were examined for viral replication compartment formation efficiency at 6 hpi (*Xu and Roizman, 2017*). As shown in *Figure 3G*, all stages of replication compartments, delegated by aggregation of the viral single-stranded DNA binding protein ICP8 (*Xu and Roizman, 2017*), were detected in both HSV-1 genome-edged and naturally dispersed groups.

## Early accumulation of ICP0 is a contributing but not necessary condition of 'Escaping'

To obtain a more integrative picture, the expression patterns of viral genes and proteins were examined in the 'Escaping' event. For the sake of simplicity, 'Electroporation of HSV-1 sgRNA at 1 hpi during HSV-1 infection of dCas9-emerin cells' is abbreviated as 'E1hpi'. Compared with ctrl sgRNA electroporated cells, E1hpi led to significantly boosted transcription of representative viral genes starting from 3 to 4 hpi (*Figure 4A*, *Figure 4—source data 1*) and elevated protein synthesis of ICP0, ICP8, TK, and VP16 at 8 hpi and 12 hpi (*Figure 4B*, *Figure 4—source data 2 and 3*). Previously, we showed that replication compartments were formed in cells electroporated with ctrl sgRNA and HSV-1 sgRNA without observable differences. In this section, we inhibited HSV-1 DNA synthesis by phosphonoacetic acid (PAA) to examine the role of genome replication in 'Escaping' (*Godowski and Knipe, 1986*). While PAA treatment successfully inhibited viral DNA synthesis (*Figure 4—figure supplement 1A*, *Figure 4—figure supplement 1—source data 1*) and reduced the overall viral transcription level, it did not affect 'Escaping' (*Figure 4C*, *Figure 4—source data 1*), indicating that 'Escaping' involves prior molecular events before replication that led to enhanced viral gene transcription. A closer look at the early accumulation of viral proteins, including ICP0, ICP4 and ICP8, revealed that as early as

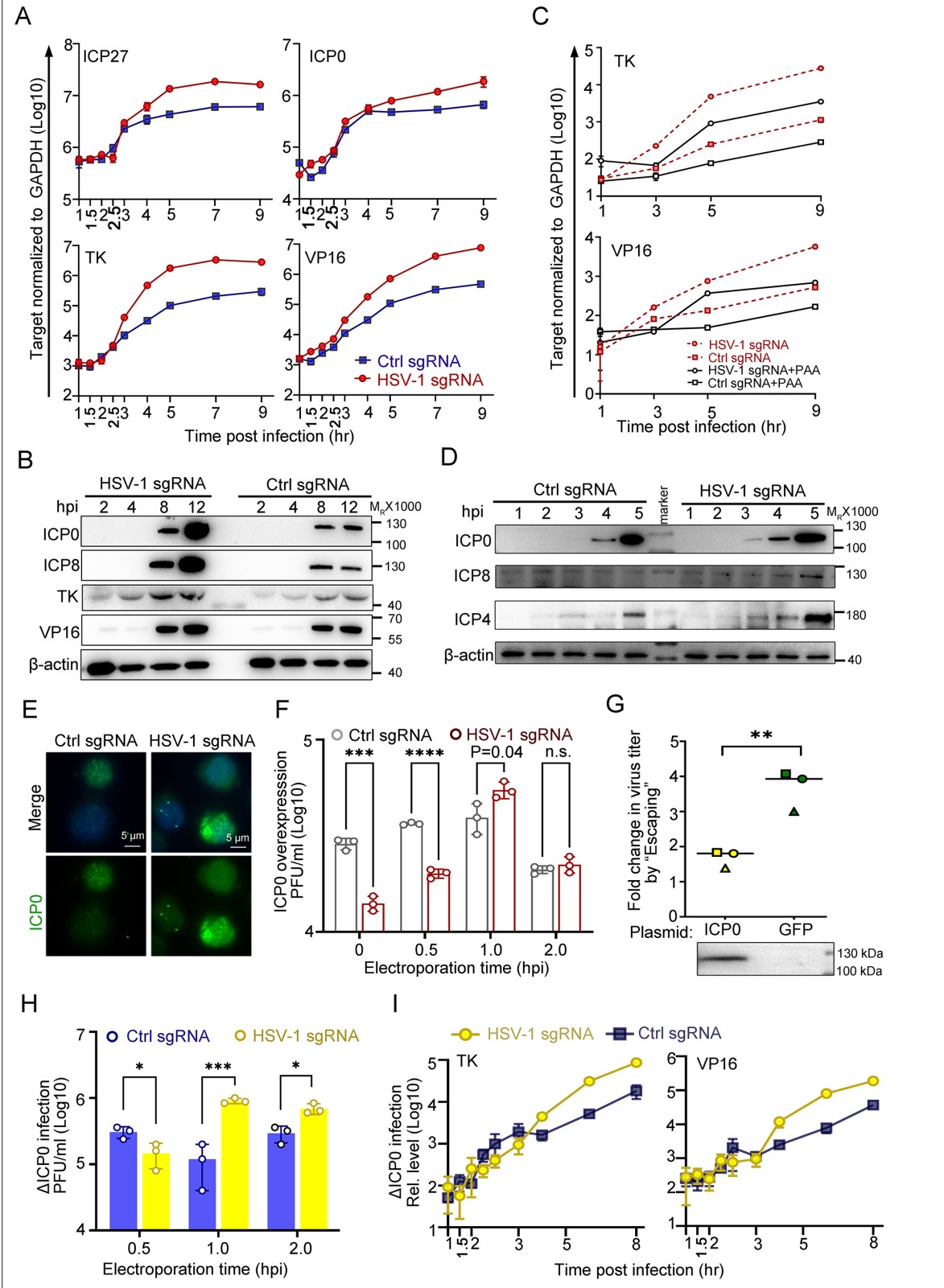

**Figure 4.** ICP0 is a sufficient but not necessary condition of the 'Escaping'. (**A–E**) dCas9-emerin cells infected with HSV-1 (2 hr on ice) at an MOI of 1 were electroporated with HSV-1 sgRNA or ctrl sgRNA at 1 hpi. (**A**) The mRNA levels of ICP27, TK, ICP0 and VP16 at the indicated time points were measured by qPCR. (**B**) The protein levels of ICP0, ICP8, TK and VP16 at the indicated time points were measured by immunoblotting. Data is shown as mean ± SD, n = 3. (**C**) dCas9-emerin cells were pretreated with PAA (500 µg/mL) or DMSO for 2 hr before infection and throughout the experiment and

*Figure 4 continued on next page*

*Figure 4 continued*

electroporated with HSV-1 sgRNA or ctrl sgRNA at 1 hpi. The mRNA levels of TK and VP16 at the indicated time points were measured by qPCR. Data is shown as mean ± SD, n = 3. (**D**) Five-fold more protein than **B** was loaded to detect early expression of viral genes by immunoblotting. (**E**) Cells were fixed at 1.5 hpi and stained with an anti-ICP0 antibody. (**F**) dCas9-emerin cells overexpressing ICP0 were infected with HSV-1 (2 hr on ice) at an MOI of 1 and electroporated with HSV-1 sgRNA or ctrl sgRNA at the indicated time points. The virus yields at 24 hpi were titrated. Data is shown as mean ± SD, n = 3, and P values are calculated using the Student's t-test. n.s. represents not significant, p>0.05, '*' represents p≤0.05, '**' represents p≤0.01, '***' represents ≤0.001, and '****' represents p≤0.0001. (**G**) dCas9-emerin cells overexpressing either ICP0 or GFP were processed as in **A**. The fold change of virus titer induced by E1hpi in ICP0 overexpressing cells versus that in GFP overexpressing cells from three independent experiments were calculated by normalizing the virus titer in cells electroporated with HSV-1 sgRNA at 1 hpi to that in cells electroporated with control sgRNA at 1 hpi and plotted (same shape of symbols represents the same batch of experiments). Data is shown as mean ± SD, n = 3, and P values are calculated using the Student's t-test. '**' represents p≤0.01. Expression of ICP0 was confirmed by immunoblotting. (**H**) Cells infected with ΔICP0 virus were processed similarly to **F**. Data is shown as mean ± SD, n = 3, and P values are calculated using the Student's t-test. '*' represents p≤0.05 and '***' represents ≤0.001. (**I**) Cells infected with ΔICP0 virus were processed similarly to (**A**) and the mRNA levels of TK and VP16 at the indicated time points were measured by qPCR. Data is shown as mean ± SD, n = 3.

The online version of this article includes the following source data and figure supplement(s) for figure 4:

**Source data 1.** Source files for western blot, statiscic data and images in *Figure 4*.

**Source data 2.** Source files for uncropped western blot in *Figure 4B*.

**Source data 3.** Source files for uncropped western blot with labels in *Figure 4B*.

**Source data 4.** Source files for uncropped western blot in *Figure 4D*.

**Source data 5.** Source files for uncropped western blot with labels in *Figure 4D*.

**Source data 6.** Source files for processed images in *Figure 4E*.

**Source data 7.** Source files for uncropped western blot in *Figure 4G*.

**Source data 8.** Source files for uncropped western blot with labels in *Figure 4G*.

**Figure supplement 1.** Characterization of the growth of ΔICP0 in dCas9-emerin cells with viral genomes inserted into the nucleus upon their entry.

**Figure supplement 1—source data 1.** Source files for western blot, statiscic data and images in *Figure 4—figure supplement 1*.

**Figure supplement 1—source data 2.** Source files for uncropped western blot in *Figure 4—figure supplement 1F*.

**Figure supplement 1—source data 3.** Source files for uncropped western blot with labels in *Figure 4—figure supplement 1F*.

3 hpi, more accumulation of ICP0 was detected in the E1hpi group (*Figure 4D*, *Figure 4—source data 4 and 5*). The immunofluorescence staining results of E1hpi cells at 1.5 hpi with anti-ICP0 antibody agreed with the observation (*Figure 4E*, *Figure 4—source data 6*). In summary, we found that dislodging viral genomes from their previously occupied niches to the nuclear fringe during a specific time window amid infection (1–1.5 hpi) led to boosted viral transcription and early accumulation of ICP0 and ICP4.

Although both ICP0 and ICP4 are critical factors in initiating the transcription of post-α genes, their functioning mechanisms are distinct. While ICP4 specifically activates viral β and γ genes and inhibits ICP0 and its own expression, ICP0 is a promiscuous transactivator (*Roizman et al., 2005*; *Watson and Clements, 1980*; *Grondin and DeLuca, 2000*; *Everett, 1984*; *Michael and Roizman, 1993*; *Bohenzky et al., 1993*). The fact that all classes of viral genes are simultaneously activated in 'Escaping' led to our further investigation of the necessity and sufficiency of ICP0 in this event. Although overexpression of the ICP0 protein mitigated the intranuclear spatial bias of HSV-1, placing incoming viral genomes to the nuclear edge still posed a significant disadvantage to virus growth (*Figure 3F*, *Figure 3—source data 1* and *Figure 4F*, *Figure 4—source data 1*). Importantly, 'Escaping' in ICP0-overexpressing cells mediated only a modest boost in virus yield (p=0.04; *Figure 4F and G*, *Figure 4—source data 1; 7 and 8*). To quantify the 'Escaping' effect in the presence or absence of overexpressed ICP0, E1hpi was performed in dcas9-emerin cells pretransfected with a plasmid expressing ICP0 or GFP. The virus titer induction level by E1hpi from three independent experiments was summarized and plotted. It is evident that overexpression of ICP0 before infection significantly reduced the 'Escaping' effect, suggesting that ICP0 accumulation early in infection contributed to this phenomenon (*Figure 4G*, *Figure 4—source data 1; 7 and 8*) individual experimental results in *Figure 4—figure supplement 1B Figure 4—figure supplement 1—source data 1*. To test whether ICP0 was also a necessary condition for 'Escaping', a mutant HSV-1 virus lacking ICP0 (ΔICP0) was used for infection. ΔICP0 virus was similarly susceptible to the nuclear edging-mediated suppression of incoming viral genomes (*Figure 4—figure supplement 1C-F*, *Figure 4—figure supplement*

*1—source data 1–3*). Furthermore, 'Escaping' significantly facilitated viral growth and promoted transcription of viral β and γ genes (*Figure 4H,I*, *Figure 4—source data 1*), and pretransfection of ICP0 before infection also dampened 'Escaping' in ΔICP0 infected cells, suggesting that ICP0 is not a necessity during 'Escaping' (*Figure 4—figure supplement 1G*, *Figure 4—figure supplement 1— source data 1*). In summary, increased accumulation of ICP0 early in infection was a contributing factor but not a necessary condition for 'Escaping'.

## Discussion

### CRISPR-nuPin is an inducible two-component platform that mediates rapid repositioning of extrachromosomal DNAs to the nuclear edge

Previous CRISPR-based genome re-organization systems use chemical- or light-inducible approaches to mediate spatial genome reorganization/engineering, and the average reported response time for these systems to relocate genomic DNA to the nuclear periphery is in the range of hours (*Morgan et al., 2017*; *Wang et al., 2018*; *Kim et al., 2019*; *Lin et al., 2019*; *Shin et al., 2019*). Our two-component CRISPR-nuPin strategy allowed us to rapidly relocate extrachromosomal DNAs up to 170 kb in minutes, reduce the time lag resulting from chemical addition and/or diffusion, and manipulate subnuclear localization of viral genomes in the middle of infection **in** minutes. CRISPR-nuPin utilized a dCas9-emerin fusion protein to tether targeted DNAs to the inner nuclear envelope upon electroporation of sgRNAs. CRISPR-nuPin utilizes a dCas9-emerin fusion protein to tether targeted DNAs to the inner nuclear envelope upon electroporation of sgRNAs and is advantageous in several aspects: (I) CRISPR-nuPin is induced through electroporation of sgRNAs and mediates prompt repositioning of targeted DNAs to the nuclear edge. Theoretically, this two-factor strategy is applicable to varying subcellular docking sites. (II) CRISPR-nuPin operates in live cells and works efficiently and fleetly during active virus infection, making it a valuable tool to interrogate the complex virus-host interactions in space and time. (III) CRISPR-nuPin relocated the viral genome up to 150 kb without the requirement of repetitive targeting sites (*Figure 2—figure supplement 1C*) and thus, in principle, could be applied to a wide range of DNA viruses, providing a method of probing subcellular viral infections spatially and temporally that has never been seen before. The study herein represents the usage of CRISPR-nuPin to synchronize and investigate the 3D locations and virological fates of HSV-1 genomes.

### Nuclear periphery: an edge of darkness for DNA viruses?

The nuclear periphery is conventionally considered a region of inactive chromatin (the B compartments), and natural nuclear envelope association frequently correlates with transcriptional repression of endogenous genes (*Dekker et al., 2013*; *Dixon et al., 2012*; *Lieberman-Aiden et al., 2009*; *Dekker et al., 2002*). However, artificially tethering otherwise located genome sites to the inner nuclear membrane showed moderate or no transcriptional suppression effects (*Wang et al., 2018*; *Reddy et al., 2008*; *Kumaran and Spector, 2008*). This inconsistent sensitivity to the nuclear periphery implies that complex mechanisms regulate the transcription of a certain endogenous gene in addition to its physical positions in the nucleus (*Reddy et al., 2008*; *Wang et al., 2018*; *Kumaran et al., 2008*; *Buchwalter et al., 2019*; *Robinett et al., 1996*; *Finlan et al., 2008*; *Zullo et al., 2012*; *Ruault et al., 2008*; *Janicki et al., 2004*). During HSV-1 infection, approximately 25–30% of viral genomes localize at the nuclear edge in HEp-2 cells (*Figure 2—figure supplement 1A*, *Figure 2—figure supplement 1—source data 1*). There is then a tantalizing question of whether the nuclear edge disfavours transcription of the incoming viral DNA compared with the inner nuclear space.

As observed in this study and other prior investigations, HSV-1 genes were transcribed at a basal level before the full onset of α gene expression (*Godowski and Knipe, 1986*). In addition, in the case of HSV-1 infecting HEp-2 cells at an MOI of 5, the basal-level transcription lasted approximately 30 min post inoculation (*Figure 2D*, *Figure 2—source data 1*). Tethering viral genomes to the inner nuclear envelope before or early (up to 0.5 hr at an MOI of 1) in this time window resulted in increased association with repressive histones, stronger transcriptional suppression, and prominently reduced virus yields. This provided the first experimental evidence showing that the nuclear edge imposes a more stringent transcription silencing on incoming viral DNA and that viral genomes entering deeper

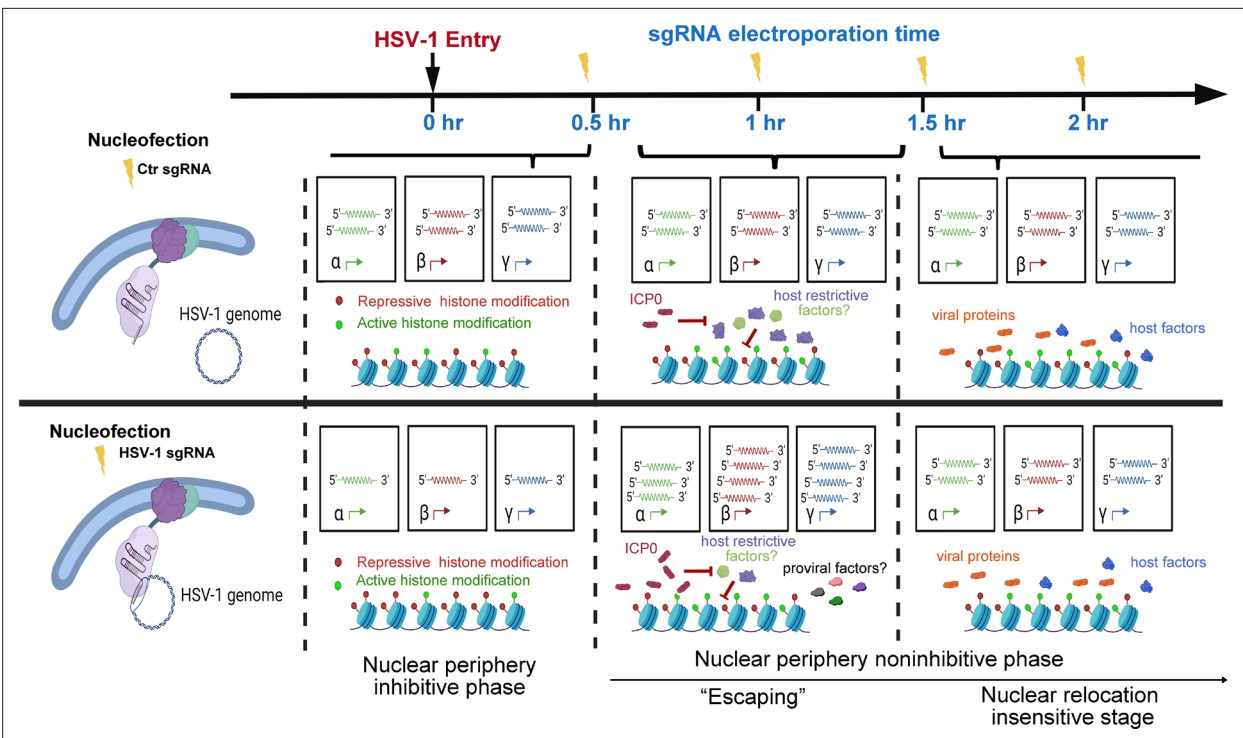

**Figure 5.** Proposed stages of the early intranuclear life of HSV-1. The schematic depicts the effects of intranuclear repositioning of viral genomes on HSV-1 infection within the initial hours of entering eukaryotic cells.

into the nucleus can be more efficiently transcribed and replicated. We thus referred to this phase as the nuclear periphery sensitive phase (*Figure 5*).

Prior studies have shown that HSV-1 tends to be targeted to the nuclear lamina and relies on A-type lamins for efficient replication and to avoid heterochromatin packaging and that HSV-1 growth was significantly compromised in lamina-deficient MEF cells (*Silva et al., 2012*; *Silva et al., 2008*; *Everett and Murray, 2005*). Despite the fact that the A-type lamin-deficient mouse embryo fibroblasts used in previous studies may have inherent defects concerning nuclear envelope integrity, genome organization, heterochromatin distribution, etc. (*Sullivan et al., 1999*; *Nikolova et al., 2004*), it is evident that it participates in the nuclear importation and organization of VP16-induced complex (VIC) formation and facilitates the transcriptional activation of viral α genes. Herein, we found that tagging HSV-1 genomes upon entry or during the first 30 min in infection to the inner nuclear membrane significantly inhibits virus transcription, including that of α genes. It should be noted that the exact meanings of 'the nuclear edge' and 'the nuclear periphery' in these studies are different at the molecular level, although they may be measured similarly under a microscope. In the CRISPR-nuPin platform, HSV-1 genomes are led directly to the surface of the inner nuclear membrane by the fusion dcas9-emerin, which likely overlays with or is embedded within the nuclear lamina. These observed inconsistencies raise a series of interesting and scientifically significant questions, starting with whether tagging HSV-1 DNAs to the inner nuclear membrane is fundamentally different from bringing them close to the lamins and why. HSV-1 genomes are transported to the nuclear pore complex in the virion and ejected into the nucleus using a pressure-driven mechanism (*Brandariz-Nuñez et al., 2019*), which is independent and distinct from the transportation and importation of VIC components VP16 and HCF-1. However, their exact difference in space and time and the molecular mechanisms mediating their encounters remain unclear. Further investigations regarding how and when the nuclear lamina helps to increase the chance of encounter between HSV-1 DNA and VIC and whether nuclear inner membrane tagging of viral genomes opposes it may shed light on this enigma. In these studies, particular attention should be paid to the timeframe post infection and the infection conditions.

## Spatial-temporal interactions between HSV-1 and host factors: fierce competition with heterogeneity

In HEp-2 cells infected with HSV-1 at an MOI of 1, HSV-1 genomes were poised for efficient infection and no longer susceptible to nuclear peripheral positioning-mediated suppression after 0.5 hpi. We referred to this period as the second phase of early infection—the nuclear periphery noninhibitive phase (*Figure 5*). Interestingly, there was a short time window (0.5–1.5 hpi) when repositioning HSV-1 genomes away from their previously occupied slots by CRISPR-nuPin to the nuclear rim significantly promoted viral transcription in a replication-independent mechanism. This effect could be due to either or a combination of the two sequential events: removal of viral genomes from their prior niches and relocation of viral genomes to the nuclear fringe.

One possible hypothesis that fits into this scenario is that during this phase (0.5–1.5 hpi at an MOI of 1), inhibitory factors are flooding HSV-1 genomes to restrict viral transcription, which is counteracted by ICP0, and therefore, HSV-1 genomes being relocalized intranuclearly helps the virus to escape from the incoming restriction ('Escaping'). If this hypothesis holds true, enhancing the mobility of HSV-1 genomes during this phase or relocating viral genomes to any other intranuclear positions should result in a comparable 'Escaping' effect. Another possible explanation for this phenomenon is that repositioned viral genomes gain greater access to pro-viral factors at the nuclear boundary. Should the second scenario prove accurate, the 'Escaping' effect would only occur when viral genomes are relocated close to the inner nuclear membrane at 1 hpi.

At 2 hpi in HEp-2 cells exposed to HSV-1 at an MOI of 1, repositioning of viral genomes within the nucleus ceased to have a discernible effect on the ultimate virus yield. This observation suggests that a position-sensitive phase of host-virus competition concluded at this specific time point (*Figure 5*).

In summary, this example study using CRISPR-nuPin in HSV-1 early life in the nucleus has shown that not only do viral genomes entering the nucleus confront heterogeneous microenvironments but relocating viral genomes to the nuclear rim during a specific time period in early infection helps the virus gain growth advantages.

## Materials and methods

### Plasmids and Transfection

The emerin coding sequence was amplified from the cDNA of HEp-2 cells. dCas9 (D10A/N863A) was amplified from the lentiSAMv2 backbone (Addgene, #75112). The coding sequences of dCas9 and emerin were inserted into a pcDNA3.1 vector and arranged as NLS (nuclear localization signal)-dCas9-NLS-emerin-Flag-T2A-GFP under a CMV promoter for transient transfection. The NLS-dCas9-NLS-emerin-Flag-T2A-GFP was cloned into the all-in-one doxycycline inducible lentiviral vector pCW57-MCS1-2A-MCS2, a gift from Pro. Deyin Guo's lab (Sun Yat-sen University) for lentivirus packaging and transduction of HEp-2 cells. The 170 kb plasmid used in *Figure 1* was a bacterial artificial chromosome containing the wild type HSV-1 (F strain) genomic sequence with the BAC vector inserted in the TK gene of HSV-1 (BAC) (*Horsburgh et al., 1999*). The 5 kb plasmid used in *Figure 1—figure supplement 1* was a modified pcDNA3.1 vector. All constructs were verified by sequencing.

Transient transfection of HEK293T cells used standard polyethylenimine (PEI) (Sigma, #408727). For transfection of HEp-2 cells and other mammalian cell lines, Jetprime (Polyplus, #101000046) was used according to the manufacturer's protocol.

### Cells, viruses, and infection

HEp-2, HEK293T and Vero cells were grown in Dulbecco's modified Eagle's medium (DMEM, Corning, #10013074) supplemented with 10% fetal bovine serum (FBS, Gibco, #42G4086K). U2OS cells were cultured in McCoy's 5 A (Procell, #PM150710) with 10% FBS. dCas9-emerin cells were established by lentivirus transduction of HEp-2 cells, selected under puromycin (1 µg/ml) (Gibco, #A1113803), induced by doxycycline (DOX) at 3 µg/ml and sorted for high GFP-expressing cells by MoFLO Astrios EQs (Beckman Coulter Life Sciences) using 488 nm lasers. dCas9-emerin cells were treated with DOX for at least 3 days before HSV-1 infection experiments. HEp-2 cell line was a kind gift from Prof. Guoying Zhou's lab at Guangzhou Medical University, HEK293T and Vero cells were gifts from Deyin Guo's lab at Sun Yat-sen University. U2OS cell line was commercially purchased (CL-0236, Procell).

All cells are originally authenticated with STR profiling and routinely tested negative for mycoplasma contamination.

HSV-1(F) and recombinant HSV-1 carrying EGFP gene fused to the C terminal of ICP0 (R8515) virus were prepared in HEp-2 cells and titrated in Vero cells by plaque assay (*Gu et al., 2009*). Recombinant HSV-1 lacking the ICP0 gene (ΔICP0),initially constructed in Dr. Roizman's lab (R7910), was amplified, and titrated in U2OS cells by plaque assay (*Kawaguchi et al., 1998*). R7910 has genomic deletions of two regions: AseI (1564–1569) to NotI (7881–7888) and NotI (118310–118317) to AseI (124629–124634). Both viruses contain two copies of HSV-1 sgRNA2 targeting site as HSV-1 (F). Viruses were stored at –80°C as single-use aliquots.

Infections performed by inoculating cells with HSV-1 virus at the desired MOI on ice for 2 hr are noted in the figure legends. The remaining unnoted virus inoculations, including plaque assays, virus amplification, growth kinetics, etc., were performed at 37 °C for 2 hr. Immediately post 2 hr inoculation was referred to as 0 hpi. Infected cells were cultured in DMEM supplemented with 1% FBS at 37 °C. At the indicated time points, the HSV-1 titer was determined by collecting all infected cells, freezing and thawing three times, sonicating them at 20% amp for 7 s and titrating them into Vero cells.

## Reagents
Phosphonoacetic acid (Sigma, #284270), valproic acid (MedChemExpress, #HY-10585), romidepsin (MedChemExpress, HY-15149), sodium butyrate (Vetec, #V900464), trichostatin A (Selleck, #S1045), DMSO (Merck, #D2650).

## Lentivirus production
To produce lentivirus, HEK293T cells were seeded in 60 mm dishes and transfected with packaging plasmids pCMV-dR8.91 and VSV-G and a lentiviral backbone plasmid containing the dCas9-emerin-flag-T2A-GFP insertion or GFP insertion only at an appropriate ratio. The medium was changed at 6–8 hr post transfection. Culturing medium was collected at 48 hr post transfection, filtered through 0.45 µm filters and centrifuged at 1000 $g$ for 10 min to remove cell debris. The supernatant containing lentiviruses was added freshly to cells for infection or frozen at –80°C.

## Fluorescence in situ hybridization (FISH)
HSV-1 DNA FISH probes were prepared from BAC-HSV-1 through nick translation (exon biotechnology, #21076) with Cy3-labeled dNTPs. Then, 5 kb plasmid DNA FISH probes were prepared similarly from the plasmid through nick translation. Briefly, 2 µg of template DNA was mixed with Cy3-labeled dNTPs, nick translation buffer, nuclease-free water, and nick translation enzyme according to the manufacturer's protocol. Then, the mixture was incubated for 1–5 hr at 15°C. The labeled probe was precipitated and redissolved in nuclease-free water. The telomere FISH probe was purchased from PNA (TelG-Alexa488, #F1008). The hybridization procedure was slightly modified from a previously described protocol (*Kawaguchi et al., 1998*; *Bayani and Squire, 2004*). In brief, the cells were firstly fixed with 4% paraformaldehyde (Sigma, #P6148) at room temperature (RT) for 15 min. For telomere staining, cells were additionally incubated with 0.1% Tris-HCl pH 7.0 for 10 min, permeabilized with 0.8% Triton (Sigma, #T8787) in PBS at RT for 10 min, incubated with 20% glycerol (Sangon Biotech, #A600232-0500) in PBS for 20 min, and permeabilized with 0.8% Triton in PBS for 30 min before RNase A treatment. For hybridization of plasmids, BAC DNAs and HSV-1 genomes, cells were directly treated with RNase A (Omega, #L15UM) at 37 °C for 50 min, and incubated with 50% formamide (VETEC, #V900064) in 2 X saline-sodium citrate (SSC) buffer (0.3 M NaCl, 30 mM sodium citrate) for 10 min, with 2XSSC washes between steps. The cells were then treated with 75%, 85%, and 100% ethanol gradients each for 2 min. They were incubated with fluorescence labeled probes diluted to 20 µg/ml in hybridization solution (2XSSC, 50% formamide, 10% dextran sulfate (Sigma, #D8906), 1% Triton) at 85 °C for 10 min and 37 °C for 20 hr. After hybridization, the cells were sequentially washed with PBS containing 75%, 50% and 25% wash buffer (2XSSC, 70% formamide, 10% dextran sulfate).

For the FISH analysis of electroporated cells, a sequential procedure was followed. Initially, the cells were fixed in a 4% PFA solution for 15 min at RT in Eppendorf tubes. Subsequently, the fixed cells were pelleted by centrifugation at 800 $g$ for 5 min, resuspended in 10–20 µl of DMEM, and carefully seeded onto lysine-coated micro-slides. The slides were then allowed to naturally air dry, facilitating proper attachment of the cells, and rendering them ready for the FISH procedure.

For FISH and immunofluorescent staining, cells were further processed as described in the immunofluorescent staining section.

## Quantitative real-time PCR

Total RNA was extracted using an E.Z.N. A Total RNA Kit I (OMEGA, #R6834-02), followed by cDNA synthesis using the Evo M-MLV RT Kit (Accurate biology, #AG11603). Total DNA was isolated using a Virus DNA Kit (OMEGA, #D3892-01). qPCR was performed using the Sybrgreen detection systems (Accurate biology, #AG11701) on the StepOneTM system (Thermo Fisher). For quantification of HSV-1 gene expression, RNA level of viral genes was normalized to GAPDH. Primers used in qPCR

| Genes | Sence | Antisence |
| --- | --- | --- |
| ICP0 | ACTGCCTGCCCATCCTGGACA | CCATGTTTCCCGTCTGGTCC |
| ICP27 | CGGGCCTGATCGAAATCCTA | GACACGACTCGAACACTCCT |
| TK | CCAAAGAGGTGCGGGAGTTT | CTTAACAGCGTCAACAGCGTGCCG |
| VP16 | CCATTCCACCACATCGCT | GAGGATTTGTTTTCGGCGTT |
| Us1 | TCGGCAGTATCCCATCAGGT | TCGGCAGTATCCCATCAGGT |
| Us12 | AACGCACCAAACAGATGCAG | CGTCCAAACCCACCGACATA |
| GAPDH | GAAGGTGAAGGTCGGAGTC | GAAGATGGTGATGGGATTTC |
| ICP27 promoter | CCGCCGGCCTGGATGTGACG | CGTGGTGGCCGGGGTGGTGCTC |
| ICP8 promoter | CCACGCCCACCGGCTGATGAC | TGCTTACGGTCAGGTGCTCCG |
| VP16 promoter | GCCGCCCCGTACCTCGTGAC | CAGCCCGCTCCGCTTCTCG |
| GAPDH promoter | TTCGACAGTCAGCCGCATCTTCTT | CAGGCGCCCAATACGACCAAATC |

## Cell viability assay

Cell viability assays were performed using a Cell Counting Kit-8 (CCK-8, Bimake, #B34302) according to the manufacturer's instructions. The fluorescence intensity was measured in a Synergy H1 microplate reader (Biotek) at 450 nm. Wells containing only culturing medium served as controls.

## Subcellular fractionation

Subcellular fractionation was performed as previously described (*Xu and Roizman, 2017*; *Ma et al., 2022*). In short, $1 \times 10^6$ cells were collected, pelleted by centrifugation at 1000 $g$ for 5 min, resuspended in 0.3 ml of ice-cold buffer 1 (150 mM NaCl, 50 mM HEPES [pH 7.4], 25 µg/ml digitonin, 10 µl/ml protease inhibitor), incubated for 30 min at 4 °C and then centrifuged at 4600 rpm for 5 min. The supernatant was collected, and the pellets were washed and resuspended in 0.3 ml of ice-cold buffer 2 (150 mM NaCl, 50 mM HEPES [pH 7.4], 1% [vol/vol] NP-40, 10 µl/ml protease inhibitor) and incubated for 30 min on ice. The samples were centrifuged at 8700 rpm for 5 min to pellet nuclei, and the supernatants were collected and combined with the previous collection, representing the cytosolic membrane-enclosed organelles. The pellets were washed and resuspended in 0.2 ml of ice-cold buffer 3 (150 mM NaCl, 50 mM HEPES [pH 7.4], 0.5% [wt/vol] sodium deoxycholate, 0.5% [wt/vol] SDS, 1 mM DTT, 10 µl/ml protease inhibitor) on ice for 30 min, followed by sonication (10 s, 20% amplification). The final solution contained the nuclear extract.

For endoplasmic reticulum membrane separation, a total of $1 \times 10^6$ cells were collected at the indicated time points post infection and pelleted down at 2000 $g$ for 5 min. One-fifth of the cells were aliquoted and served as the total control. The remaining cells were separated into subcellular compartments as described above and the supernatant collected after buffer 2 lysis representing the cytosolic membrane-enclosed organelles was collected.

## Western blots

In brief, lysates were separated on polyacrylamide gels, transferred onto PVDF membranes, and blotted with the indicated antibodies: mouse monoclonal anti-Flag antibody (Abways, #AB0008), mouse monoclonal anti-ICP0 antibody (Santa Cruz, #sc-53070), mouse monoclonal anti-ICP8 antibody

(Abcam, #ab20194), mouse monoclonal anti-VP16 antibody (Santa Cruz, #sc-7545), mouse monoclonal anti-β-actin antibody (Sino Biological, #1000166), anti-TK antibody (laboratory stock), rabbit anti-E-cadherin antibody (Affinity Biosciences, #AF0131), mouse monoclonal anti-GAPDH antibody (Abways, #AB0037), mouse monoclonal anti-Histone antibody (Sino Biological, #100005), rabbit polyclonal anti-dCas9 antibody (ABclonal, #A14997), goat anti-mouse IgG-HRP (Invitrogen, #31430), and goat anti-rabbit IgG (H+L)-HRP (Invitrogen, #32460).

## Immunofluorescence (IF) staining
Cells were fixed in methyl alcohol at –80°C overnight, then permeabilized and blocked with PBS-TBH (0.1% Triton X-100 in 1XPBS, 10% FBS, and 1% bovine serum albumin (BSA) (Sigma, #WXBD5147V)), reacted with primary antibodies overnight at 4°C or 2 hr at RT, and secondary Alexa Fluor 594-conjugated goat anti-rabbit (Invitrogen, #A11012) or Alexa Fluor 488-conjugated goat anti-mouse (Invitrogen, #A32723) for 30 min at RT in the dark. The slides were mounted with mounting medium and imaged with a Zeiss confocal microscope (ZEISS Imager Z20).

## sgRNA design
Genomic information of the HSV-1 (F) strain genome was downloaded from GenBank (Accession: GU734771). sgRNAs were designed using Cistrome (http://cistrome.org/SSC/). Ctrl sgRNA (5'-GGGG TAGGCGGAGCCTCAGG-3') has no completed matched targets in the human genome or HSV-1 genome.

## Electroporation
Cells were deattached with 0.1% trypsin, washed with DPBS, resuspended in 100 µl of electroporation buffer (Cell Line Nucleofector Kit V, #VCA1003), mixed with DNA and/or RNA as indicated in each experiment and electroporated at 110 V for 10ms by the electroporator (Lonza Amaxa Nucleofector 2B). After electroporation, the cells in each electroporation cup were mixed with 1 ml DMEM and placed back into a culture dish for subsequent experiments.

## In vitro transcription
In vitro sgRNA transcription was performed using the Ribomax T7 large-scale RNA production kit (Promega, #P1300) according to the manufacturer's instructions. In short, 2–4 µg of template DNA containing a T7 promoter was used for each reaction. The in vitro transcription mixture was incubated at 37 °C for 3.5 hr, and RNase-free DNase I was added to remove the DNA template. The transcribed RNAs were treated with alkaline phosphatase (Thermo, #01137175) for 1 hr and cleaned up using an E.Z.N.A. miRNA Kit (Omega, #R6842-01). The final RNA stocks were quantified using a NanoDrop spectrophotometer, aliquoted and stored at –80°C.

## Confocal microscopy and image processing
Confocal images of the cell samples were taken with a Nikon microscope (A1R N-SIM N-STORM) with 100 X-oil immersion lenses. The confocal images were processed using NIS Elements software by time-lapse microscopy with Z-stacks. Imaris software (Bitplane) was used to perform 3D visualization and 3D reconstitution. DNA FISH spots were built using the Spots function, and the surface of the nucleus was built as a membrane object using the cell function and then switched to a surface object. The number of plasmids or HSV-BAC on the nuclear envelope was quantified by using the reconstituted plasmids or HSV-BAC spots and the nuclear periphery surface.

In *Figure 2—figure supplement 1B*, the distance between the FISH spots and the nuclear periphery (DAPI edge) in the confocal images was determined in ImageJ software by an image ruler. In brief, the shortest distance between the center of a FISH spot and the DAPI edge was measured by the straight-line selection tool in image J. 100 nuclear-edged FISH points were measured and a frequency distribution graph was generated using GraphPad Prism.

## Chromatin immunoprecipitation (ChIP)
ChIP assays were carried out as described previously. In short, at the indicated time points postinfection, cells were treated with 1% paraformaldehyde (Sigma, #P6148) for 10 min, washed twice with PBS, scraped off, resuspended in sodium dodecyl sulfate (SDS) lysis buffer (1% SDS 10 mM EDTA

50 mM Tris, pH 8.1) containing protease inhibitors (Thermo Fisher Scientific, #EO0492), and incubated on ice for 20 min. Cell lysates were sonicated in 20 s pulses for a total of 4 min to yield DNA fragments of 200–500 bp in length and further clarified by centrifugation at 13,000 $g$ at 4 °C for 10 min. The supernatant was collected and diluted 10-fold in phosphate-buffered radioimmunoprecipitation assay buffer (0.1% SDS, 1% sodium deoxycholate, 150 mM NaCl, 10 mM $Na_2PO_4$, 2 mM EDTA, 1% NP-40) and precleared for 1 hr at 4°C. At this point, 10% of the total volume was aliquoted as input and total DNA was extracted for quantification of total viral DNA and cellular DNA input by qPCR. Immunoprecipitation was carried out at 4°C overnight with mouse immunoglobulin G (Cell Signaling Technology, 5415 S) or anti-histone antibodies (anti-H3K4me3 antibody (Abcam, #ab8580), anti-H3K9me3 antibody (Abcam, #ab8898), anti-H3K27me3 antibody (Abcam, #ab6002), anti-H4K20me3 antibody (Abcam, #ab9053), anti-H3K9AC antibody (Abcam, #ab4441), and anti-H3 antibody (Abcam, #ab1791)). Immunocomplexes were collected by incubation with protein A/G beads (Santa Cruz, #sc-2003) for 1 hr at 4°C with rotation, washed sequentially with low-salt wash buffer (150 mM NaCl, 20 mM Tris HCl, pH 8.1, 2 mM EDTA, 1% Triton X-100, and 0.1% SDS), high-salt wash buffer (500 mM NaCl, 20 mM Tris HCl, pH 8.1, 2 mM EDTA, 1% Triton X-100, and 0.1% SDS), lithium chloride wash buffer (0.25 M LiCl, 1% NP-40, 1% sodium deoxycholate, 1 mM EDTA, and 10 mM Tris-HCl, pH 8.1), and Tris-EDTA buffer (10 mM Tris-HCl, pH 8, 1 mM EDTA) and eluted by incubation with elution buffer (1% SDS, 0.1 M NaHCO3) at RT for 10 min, at 65°C for 10 min, and finally at RT for 10 min. NaCl was added to both eluates and inputs to reach a final concentration of 0.2 M. All samples were then treated with RNase A (Omega, #D10SG) and proteinase K (QIAGEN, #160016374), and total DNA was purified by E.Z.N.A. Gel Extraction Kit (Omega, #D2500-02) and used as a template for qPCR.

To measure the input amounts and percentage of immunoprecipitated viral and host DNA, quantitative PCR was performed using primers specific for viral promoters and cellular GAPDH. In *Figure 2F and G* and *Figure 2—figure supplement 1M*, the relative input amount of a particular viral promoter region was first normalized to the input GAPDH level (value X). Then, the relative level of the viral promoter region pulled down by a specific histone antibody during the CHIP assay was normalized to the GAPDH level pulled down by the same histone antibody (value Y). Finally, the fold enrichment of the certain viral promoter region on the specifically modified histone was calculated by dividing Y by X (value Y/X). *Figure 2F and G* show the plotted value Y/X with the Y axis displaying Fold enrichment/GAPDH (Log10). An example of the fold enrichment of ICP27 promoter region on H3K4me3 is shown in the equation below:

$$\text{Fold enrichment/GAPDH} = \frac{Relative \frac{ICP27}{GAPDH} \; by \; CHIP \; using \; H3K4me3 \; antibody}{Input \; relative \; ICP27/GPADH}$$

## Statistical analysis

qPCR CHIP-qPCR, growth curve was performed with aliquots from the same sample. Key experiments (*Figure 1F and H*, *Figure 2A–H*, *Figure 3E and F*, and *Figure 4A–I*) were repeated three times or more to ensure reproducibility. For *Figure 1F and H*, *Figure 3D* and *Figure 1—figure supplement 1* I, 100–300 spots were counted at a time and 4–5 sets of spot counting were conducted on different images from the same experimental group, resulting in 4 or 5 values that gave the error bar. Data are presented as the mean ± sd. calculated by GraphPad Prism 6.0 software. Two-tailed unpaired Student's t-test, ordinary one-way or two-way ANOVA indicated in each figure were used to calculate p values. n.s. represents not significant, $p > 0.05$, '*' represents $p \leq 0.05$, '**' represents $p \leq 0.01$, '***' represents '$\leq 0.001$, and '****' represents $p \leq 0.0001$.

## Materials availability statement

All newly created materials, including plasmids and cell lines, are available upon request to the corresponding author. All data for charts and tables, original gels and blots were provided as source data as indicated.

## Acknowledgements

This project is supported by the National Key R&D Program of China (2022YFC2305400); the National Natural Science Foundation of China (No. 31870157); the Shenzhen Science and Technology Innovation

Program (JCYJ20180307151536743 and KQTD20180411143323605) and Shenzhen Natural Science Foundation (JCYJ20220530145810023).

## Additional information

### Funding

| Funder | Grant reference number | Author |
|---|---|---|
| National Key Research and Development Program of China | 2022YFC2305400 | Pei Xu |
| National Natural Science Foundation of China | No. 31870157 | Pei Xu |
| Shenzhen Science and Technology Innovation Program | JCYJ20180307151536743 | Pei Xu |
| Shenzhen Science and Technology Innovation Program | KQTD20180411143323605 | Pei Xu |
| Natural Science Foundation of Shenzhen Municipality | JCYJ20220530145810023 | Pei Xu |

The funders had no role in study design, data collection and interpretation, or the decision to submit the work for publication.

### Author contributions

Juan Xiang, Data curation, Software, Validation, Investigation, Took the lead in conducting the majority of FISH and qPCR experiments, worked closely with ZYF on various aspects of the research, contributed to manuscript preparation, and provided assistance in the review and editing process; Chaoyang Fan, Validation, Investigation, Initiated and developed the dCas9-emerin platform, collaborated extensively with JX on various experiments, and provided support in preparing the manuscript; Hongchang Dong, Investigation, Provided assistance in performing electroporation and western blotting experiments; Yilei Ma, Investigation, Provided assistance with the western blotting experiment; Pei Xu, Conceptualization, Resources, Formal analysis, Supervision, Funding acquisition, Methodology, Writing - original draft, Project administration, Writing - review and editing

### Author ORCIDs

Juan Xiang (ID) https://orcid.org/0000-0003-3853-7762
Chaoyang Fan (ID) https://orcid.org/0000-0003-2286-4865
Hongchang Dong (ID) https://orcid.org/0000-0002-8190-2158
Pei Xu (ID) https://orcid.org/0000-0003-2719-5605

### Decision letter and Author response

Decision letter https://doi.org/10.7554/eLife.85412.sa1
Author response https://doi.org/10.7554/eLife.85412.sa2

## Additional files

### Supplementary files

• Supplementary file 1. Repeat numbers of key experiments. The contents of this Excel file provide information about the repeat numbers of crucial experiments conducted during the research.

• MDAR checklist

### Data availability

No datasets and code were generated or used during the study. All raw microscopy images, original files for charts and blots are available on Dryad at https://doi.org/10.5061/dryad.vmcvdncxd.

The following dataset was generated:

| Author(s) | Year | Dataset title | Dataset URL | Database and Identifier |
|---|---|---|---|---|
| Xu P | 2023 | A CRISPR-based rapid DNA repositioning strategy and the early intranuclear life of HSV-1 | https://doi.org/10.5061/dryad.vmcvdncxd | Dryad Digital Repository, 10.5061/dryad.vmcvdncxd |

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
