## [Editor Report]

The authors present an ingenious approach to investigate the effects of cellular location on herpes virus replication. By infecting cells that express a fusion protein between Cas9 and emerin, a nuclear matrix protein, the addition of guide RNAs that target the herpes virus DNA genome quickly localize it to the nuclear membrane. Initial inhibition of viral gene expression, followed by release of that inhibition in just one hour, was observed. This is an interesting approach to test the importance of localization within the nucleus that is reveals both an effect of 'tethering' and a pathway for escape of that tethering, using a method relevant to other viral and non-viral nucleic acids.

---

## [Decision Letter]

**Decision letter after peer review:**

Thank you for submitting your article "A CRISPR-based instant DNA repositioning strategy and the early intranuclear life of HSV-1" for consideration by *eLife*. Your article has been reviewed by 2 peer reviewers, one of whom is a member of our board of Reviewing Editors, and the evaluation has been overseen by Kevin Struhl as the Senior Editor. The reviewers have opted to remain anonymous.

Essential revisions:

1) Rewriting the interpretations of the timing of DNA relocalization. Repeatedly there are statements of "instant" relocalization, but these are based on the observations of shifted distributions at the "10 second" samples. From the written methods it appears that 4% PFA fixation was employed at RT for 15 minutes. PFA is not an instantaneous fixative that locks components in place. A great article detailing some of the assumptions of fixation is Schnell U et al. Immunolabeling artifacts and the need for live-cell imaging. Nat Meth. 2012 Jan 30;9(2):152-8. The data support an interpretation of rapid relocalization within minutes of application.

2) Interpretation of DNA localization data and description of imaging methodology. The images and 3D reconstruction are striking, but there are critical details missing from the methods that directly influence the interpretation of the data. Line 689 claims confocal microscopy, but then states an N-STORM microscope was used. If these images were acquired with super-resolution methods, then those methods should be detailed. If these are standard confocal images, then the measurements provided in the results (lines 203-207) are imprecise. Theoretically a confocal can achieve xy-resolution between 250 – 400 nm, depending on acquisition parameters. Superresolution could achieve the resolutions reported.

3) The employ of euphemism and evocative language detracts from the explanation and interpretation of results. Especially, the use of driving euphemisms like "full-speed" (line 244) and "higher gear" (line 247) to convey changes in transcript measurements implies that control of transcript abundance occurs at the level of transcription initiation. Please directly stated the observed measurement is sufficient to explain that sgRNA type has an effect on the extent of RNA accumulation, with HSV-specific guides reducing target RNA detection.

4) The replicate number of experiments. Line 754-755, "Key experiments (Figure 1 F and H, Figure 2 A-H, Figure 3 E, and F, and Figure 4 A-I) were repeated three times or more to ensure reproducibility." This statement does not address the data that are displayed or the representation of error bars. How many replicates were measured for each condition in each graph? How many times were Western Blot experiments repeated? For the % distribution, how many nuclei are represented in the 'number of spots' values? The prior statement says qPCR was triplicate measured from a single sample, yet those experiments are presented in 2F and G which were repeated multiple times. Clarifications of the replicate numbers for all experiments need to be made as they relate to the data presented.

5) Additional control experiments and data presentation are needed to support the stated results. Specific examples:

a. Line 344 "were detected in both HSV-1 genome-edged and naturally dispersed groups with no distinguishable overall pattern differences." This declaration is not sufficiently supported by the presented data. How many cells were imaged to reach this conclusion? Looking at the data suggests asymmetry of ICP8 distribution. What is the frequency of each class of ICP8 distributions? What are the criteria for classification? Size of aggregate? Please either revise this statement or provide additional analysis of the data.

b. Line 369 "the immunofluorescence staining results…" The immunofluorescence data lack appropriate controls for infection. Quantitation of fluorescence intensity or classification of structures across a population of image cells needs to be presented, instead of showing a non-staining cell from the control and a stained cell from the treatment.

6) Line 499 "suggested that ICP0 was sufficient…" 3Logic throughout the manuscript needs some re-phrasing. For example, the conclusion that ICPO is 'sufficient but not necessary' for the relief of viral gene expression is difficult for this reviewer to understand. It simply seemed not to be necessary.

---

## [Author Response]

Essential revisions:1) Rewriting the interpretations of the timing of DNA relocalization. Repeatedly there are statements of "instant" relocalization, but these are based on the observations of shifted distributions at the "10 second" samples. From the written methods it appears that 4% PFA fixation was employed at RT for 15 minutes. PFA is not an instantaneous fixative that locks components in place. A great article detailing some of the assumptions of fixation is Schnell U et al. Immunolabeling artifacts and the need for live-cell imaging. Nat Meth. 2012 Jan 30;9(2):152-8. The data support an interpretation of rapid relocalization within minutes of application.

We appreciate the reviewer’s advice and have rephrased “instant” and all other interpretations of the timing of DNA relocalization to “rapid” or a similar phrasing in the manuscript.

2) Interpretation of DNA localization data and description of imaging methodology. The images and 3D reconstruction are striking, but there are critical details missing from the methods that directly influence the interpretation of the data. Line 689 claims confocal microscopy, but then states an N-STORM microscope was used. If these images were acquired with super-resolution methods, then those methods should be detailed. If these are standard confocal images, then the measurements provided in the results (lines 203-207) are imprecise. Theoretically a confocal can achieve xy-resolution between 250 – 400 nm, depending on acquisition parameters. Superresolution could achieve the resolutions reported.

Thank you for bringing the confusing descriptions to our attention. We utilized a N- STORM microscope (A1R N-SIM N-STORM), which is equipped with a high- resolution galvano (non-resonant) scanner and an ultrahigh-speed resonant scanner, for confocal imaging. The immunofluorescence and DNA FISH images used for 3D reconstruction were traditional confocal images, without the use of super-resolution methods. The revised material and method section includes details on the image and the microscope information. The updated description is as follows:

"Confocal images of the cell samples were captured using a Nikon microscope (A1R N-SIM N-STORM) equipped with 100X oil immersion lenses.” (Line 724-726).

We concur with the reviewers that the resolution limit of the confocal images is not less than 250 nm. In this study, confocal microscopy combined with 3D reconstruction is adequate for determining the location of FISH signals in relation to the peripheral zone of the nucleus (the definition of nuclear-edged versus intranuclear viral genomes is elaborated in the manuscript (Line 167-line 171)). We would also like to clarify that we do not intend to claim to have measured distances that fall below the confocal resolution. To prevent any confusion, we have made modifications to our description and numbers in the manuscript (Line 204-207).

While the confocal resolution limit is greater than 250 nm, the pixel size of the images used in this study was set between 50-100 nm. Therefore, in Sup Figure 2 B, the calculation of the distance between the center of a FISH spot and the edge of the DAPI area in these confocal images could result in numbers smaller than the confocal microscopy resolution limit if the FISH spots are located near the DAPI edge. We have included a supplement to the material and method section to explain the distance measurement method used in Figure 2—figure supplement 1 B. The updated description is as follows:

"In Figure 2—figure supplement 1 B, the distance between the FISH spots and the nuclear periphery (DAPI edge) in the confocal images was determined using ImageJ software with an image ruler. The shortest distance between the center of a FISH spot and the DAPI edge was measured by the straight-line selection tool in ImageJ. We measured 100 nuclear-edged FISH puncta and generated a frequency distribution graph using GraphPad Prism." (Line 734-739 of the revised manuscript).

3) The employ of euphemism and evocative language detracts from the explanation and interpretation of results. Especially, the use of driving euphemisms like "full-speed" (line 244) and "higher gear" (line 247) to convey changes in transcript measurements implies that control of transcript abundance occurs at the level of transcription initiation. Please directly stated the observed measurement is sufficient to explain that sgRNA type has an effect on the extent of RNA accumulation, with HSV-specific guides reducing target RNA detection.

Dear editors and reviewers, we used the terms "full-speed" (line 244) and "higher gear" (line 247) to refer to the control of transcript abundance at the level of transcription initiation. As per previous research, α genes of HSV- 1 are silenced by host factors, including heterochromatin packaging, upon nuclear entry of HSV-1 viral genomes. These genes are then transcriptionally induced by viral protein VP16, acting in concert with Oct-1 and HCF-1 through a cis-acting site in the promoter domain of α genes (J Virol. 1991 Jul; 65(7): 3504–3513. Doi: 10.1128/jvi.65.7.3504-3513.1991). We believed that increased heterochromatin packaging (Figure 2 F, G) on nuclear edged HSV-1 genomes caused a delay in the onset of full-speed transcriptional activation of HSV-1 α genes. Once α gene transcription was fully activated, the rate of transcription was similar.

We understand the concerns raised by the editors and reviewers and have rephrased the description in Line 244-247 to make it simpler and clearer.

4) The replicate number of experiments. Line 754-755, "Key experiments (Figure 1 F and H, Figure 2 A-H, Figure 3 E, and F, and Figure 4 A-I) were repeated three times or more to ensure reproducibility." This statement does not address the data that are displayed or the representation of error bars. How many replicates were measured for each condition in each graph? How many times were Western Blot experiments repeated? For the % distribution, how many nuclei are represented in the 'number of spots' values? The prior statement says qPCR was triplicate measured from a single sample, yet those experiments are presented in 2F and G which were repeated multiple times. Clarifications of the replicate numbers for all experiments need to be made as they relate to the data presented.

We repeated each experiment independently for more than three times for Figure 1 E, F, G, and H and observed consistent patterns. For publication purposes, we presented a representative dataset of one experiment for Figure 1 E-H. This dataset was counted independently by at least two individual students, who counted at least 500 spots and about 27-250 nuclei per experimental group. The total number of nuclei counted varied in different experiments due to variations in spot numbers per nucleus. Additional details have been added to the figure legends in the revised manuscript.

When counting the intranuclear localizations of the spots, the students were instructed to count 100-300 spots at a time, resulting in a Peripheral vs. total % value. We counted 4-5 sets of 100-300 spots in images from the same experimental group, resulting in 4 or 5 values that gave the error bar shown in Figure 1 F, H, and Figure 1—figure supplement 1 I. We have added this counting method in the material and methods section of the revised manuscript (Line 796-798).

All other relevant details are provided in the source data files uploaded with the manuscript.

For Figure 2 A-H, Figure 3 E-F, Figure 4 A-I, we independently repeated the experiments more than three times and obtained consistent results. The representative experiment data was presented with triplicate measures of a single sample.

We routinely performed qPCR assay using aliquoted triplicates from a single experimental sample. The qPCR experiments were independently repeated, and the results were consistent.

An excel file containing details of how many times each experiment was independently repeated is provided in our uploaded source file (Supplementary File 1: Repeat numbers of key experiments).

5) Additional control experiments and data presentation are needed to support the stated results. Specific examples:a. Line 344 "were detected in both HSV-1 genome-edged and naturally dispersed groups with no distinguishable overall pattern differences." This declaration is not sufficiently supported by the presented data. How many cells were imaged to reach this conclusion? Looking at the data suggests asymmetry of ICP8 distribution. What is the frequency of each class of ICP8 distributions? What are the criteria for classification? Size of aggregate? Please either revise this statement or provide additional analysis of the data.

Thank you for bringing this to our attention. We agree with the comment that we did not thoroughly consider this specific point. Since this is not a very relevant statement, we have removed the unsupported assumptions in the revised manuscript.

b. Line 369 "the immunofluorescence staining results…" The immunofluorescence data lack appropriate controls for infection. Quantitation of fluorescence intensity or classification of structures across a population of image cells needs to be presented, instead of showing a non-staining cell from the control and a stained cell from the treatment.

Thanks for the valuable comment. We have updated Figure 4 E with images showing more cells exhibiting positive ICP0 expression. We would like to kindly remind editors and reviewers that at this time point post infection, all virus proteins were either not yet expressed or expressed at very low levels.

It is important to note that we do not consider immunofluorescence staining as a reliable method for directly comparing or quantifying protein levels. Consequently, we determined the protein level of ICP0 using a western blot analysis, as presented in Figure 4 D.

Moreover, we wish to clarify that the purpose of Figure 4 E is not to claim any differences in intranuclear ICP0 distribution between the two compared groups. Instead, its purpose is to provide additional support for the findings presented in Figure 4 D. This is emphasized in the manuscript on Line 370- 372, where we state:

"The immunofluorescence staining results of E1hpi cells at 1.5 hpi with anti-ICP0 antibody agreed with the observation (Figure 4 E)." In response to the reviewer's inquiry, we have also conducted a repeat of the experiment depicted in Figure 4 E, utilizing viral DNA FISH to stain HSV-1 infection (panel A and Response to reviewers source data 1). The results revealed that cells positive for HSV-1 DNA FISH generally exhibited lower levels of intranuclear ICP0 staining. A.

6) Line 499 "suggested that ICP0 was sufficient…" 3Logic throughout the manuscript needs some re-phrasing. For example, the conclusion that ICPO is 'sufficient but not necessary' for the relief of viral gene expression is difficult for this reviewer to understand. It simply seemed not to be necessary.

We agree with the reviewers that the findings presented in our investigation are complex, particularly with regards to the 'Escaping' event and the potential involvement of ICP0. To avoid overinterpretation, we exercised caution in describing, concluding, and discussing the experimental results in the manuscript. Our team recognizes the need for further investigation to determine the precise host or viral factors and their molecular mechanisms of action before drawing more specific conclusions.

As per the definition of necessity and sufficiency provided by Wikipedia, “In logic and mathematics, necessity and sufficiency are terms used to describe a conditional or implicational relationship between two statements. In general, a necessary condition is one (possibly one of multiple conditions) that must be present in order for another condition to occur, while a sufficient condition is one that produces the said condition.” In line 499, by calling ICP0 a sufficient but not necessary condition in the “Escaping”, we meant that overexpression of ICP0 in cells prior to HSV-1 infection helped to diminish the “Escaping” effect, implying its possible role in the process. However, the “Escaping” effect also occurred for a mutant HSV-1 virus lacking ICP0 expression, indicating the involvement of other factors.

Nevertheless, we agree that line 499 “our preliminary data suggested that ICP0 was a sufficient but not necessary condition in this process.” created an unnecessary difficulty for the readers. Therefore, we have removed this sentence from the manuscript.

We agree with the reviewers that the findings presented in our investigation are complex, particularly with regards to the 'Escaping' event and the potential involvement of ICP0. To avoid overinterpretation, we exercised caution in describing, concluding, and discussing the experimental results in the manuscript. Our team recognizes the need for further investigation to determine the precise host or viral factors and their molecular mechanisms of action before drawing more specific conclusions.

As per the definition of necessity and sufficiency provided by Wikipedia, “In logic and mathematics, necessity and sufficiency are terms used to describe a conditional or implicational relationship between two statements. In general, a necessary condition is one (possibly one of multiple conditions) that must be present in order for another condition to occur, while a sufficient condition is one that produces the said condition.” In line 499, by calling ICP0 a sufficient but not necessary condition in the “Escaping”, we meant that overexpression of ICP0 in cells prior to HSV-1 infection helped to diminish the “Escaping” effect, implying its possible role in the process. However, the “Escaping” effect also occurred for a mutant HSV-1 virus lacking ICP0 expression, indicating the involvement of other factors.

Nevertheless, we agree that line 499 “our preliminary data suggested that ICP0 was a sufficient but not necessary condition in this process.” created an unnecessary difficulty for the readers. Therefore, we have removed this sentence from the manuscript.